# NON-PARAMETRIC STATE-SPACE MODELS: IDENTIFIABILITY, ESTIMATION AND FORECASTING

## ABSTRACT

State-space models (SSMs) provide a standard methodology for time series analysis and prediction. While recent works utilize nonlinear functions to parameterize the transition and emission processes to enhance their expressivity, the form of additive noise still limits their applicability in real-world scenarios. In this work, we propose a general formulation of SSMs with a completely non-parametric transition model and a flexible emission model which can account for sensor distortion. Besides, to deal with more general scenarios (e.g., non-stationary time series), we add a higher-level model to capture the time-varying characteristics of the process. Interestingly, we find that even though the proposed model is remarkably flexible, the latent processes are generally identifiable. Given this, we further propose the corresponding estimation procedure and make use of it for the forecasting task. Our model can recover the latent processes and their relations from observed sequential data. Accordingly, the proposed procedure can also be viewed as a method for causal representation learning. We argue that forecasting can benefit from causal representation learning, since the estimated latent variables are generally identifiable. Empirical comparisons on various datasets validate that our model could not only reliably identify the latent processes from the observed data, but also consistently outperform baselines in the forecasting task.

## 1 INTRODUCTION

Time series forecasting plays a crucial role in various automation and optimization of business processes (Petropoulos et al., 2022; Benidis et al., 2020; Lim & Zohren, 2021). State-space models (SSMs) (Durbin & Koopman, 2012) are among the most commonly-used generative forecasting models, providing a unified methodology to model dynamic behaviors of time series. Formally, given observations $\mathbf{x}_t$, they describe a dynamical system with latent processes $\mathbf{z}_t$ as:

$$\underbrace{\mathbf{z}_t = f(\mathbf{z}_{t-1}) + \epsilon_t,}_{\text{Transition}} \quad \underbrace{\mathbf{x}_t = g(\mathbf{z}_t) + \eta_t,}_{\text{Emission}} \tag{1}$$

where $\eta_t$ and $\epsilon_t$ denote the i.i.d. Gaussian measurement and process noise terms, and $f(\cdot)$ and $g(\cdot)$ are the nonlinear transition model and the nonlinear emission model, respectively. The transition model captures the latent dynamics underlying the observed data, while the emission model learns the mapping from the latent processes to the observations. Recently, more expressive and scalable deep learning architectures were leveraged for modeling nonlinear transition and emission models effectively (Fraccaro et al., 2017; Castrejon et al., 2019; Saxena et al., 2021; Tang & Matteson, 2021).

However, these SSMs are not guaranteed to recover the underlying latent processes and their relations from observations. Furthermore, stringent assumptions of additive noise terms in both transition and emission models may not hold in practice. In particular, the additive noise terms cannot capture nonlinear distortions in the observed or latent values of the variables, which might be necessarily true in real-world applications (Zhang & Hyvarinen, 2012; Yao et al., 2021), like sensor distortion and motion capture. If we directly apply SSMs with this constrained additive noise form, the model misspecification can lead to biased estimations. Second, the identification of SSMs is a very challenging task when both states and transition models are unknown. Most work so far has focused on developing efficient estimation methods. We argue that this issue should not be ignored, and it becomes more severe when nonlinear transition and emission models are implemented with deep

learning techniques. As the parameter space has increased significantly, SSMs are prone to capture spurious causal relations and strengths, and thus identifiability of SSMs is vital. Furthermore, the transition model is usually assumed to be constant across the measured time period. This stationary assumption hardly holds in many real-life problems due to the changes in dynamics. For example, the unemployment rate tends to rise much faster at the start of a recession than it drops at the beginning of a recovery (Lubik & Matthes, 2015). In this setting, SSMs should appropriately adapt to the time-varying characteristics of the latent processes to be applicable in general non-stationary scenarios.

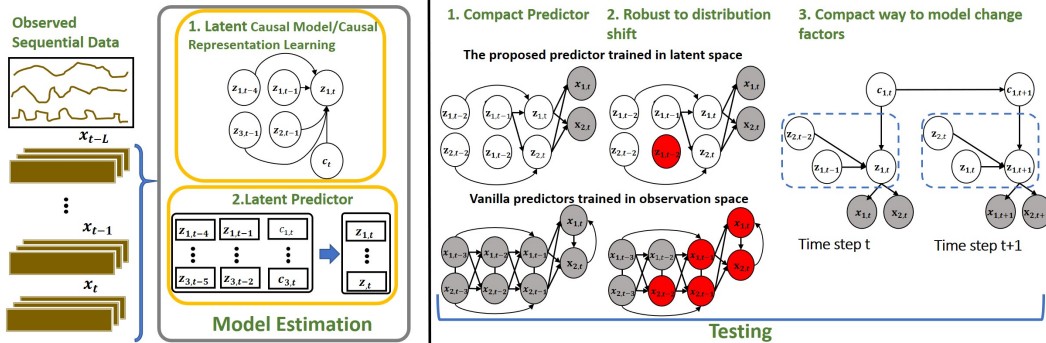

Figure 1: **Left:** The proposed estimation framework mainly includes the learning of latent causal model learning and prediction model. **Right:** Motivational examples demonstrate the benefit of latent causal model learning for forecasting. (1). It provides compact representations for forecasting, as vanilla predictors include complicated dependencies. (2). The prediction model is more robust to the distribution shift (Red circles here indicate distribution change). (3). It gives a compact way to model the change factors to address non-stationary forecasting issues.

In this work, in contrast to state-of-the-art approaches following the additive form of transition/emission models, we propose a general formulation of SSMs, called the Non-Parametric State-Space Model (NPSSM) [1]. In particular, we leverage the non-parametric functional causal model (Pearl, 2009) for the transition process and the post-nonlinear model (Zhang & Hyvarinen, 2012) to capture nonlinear distortion effects in the emission model. Besides, we add a higher level model to NPSSM, called N-NPSSM, to capture the potential time-varying change property of the latent processes for more general scenarios (e.g., non-stationary time series). Interestingly, although the proposed NPSSM is remarkably flexible, the latent processes are generally identifiable. To this end, we further develop a novel estimation framework built upon the structural variational autoencoder (VAE) for the proposed NPSSMs. It allows us to recover latent processes and their time-delayed causal relations from observed sequential data and use them to build the latent prediction model simultaneously (illustrated in Figure 1(left)). Accordingly, the proposed procedure can be viewed as a method for causal representation learning or latent causal model learning from time series data.

We argue that forecasting tasks can benefit from causal representation learning, as latent processes are generally identifiable in NPSSM. As shown in Figure 1(right), first, it provides a compact structure for forecasting, whereas vanilla predictors (bottom), which directly learn a mapping function in the observation space, face the issue of complicated and spurious dependencies. Second, the predictions following the correct causal factorization are expected to be more robust to distribution shifts that happen to some of the modules in the system. If some local intervention exists on one mechanism, it will not affect other modules, and those modules will still contribute generously to the final prediction. Although formulating this problem and providing quantitative theoretical results seem challenging, our empirical studies illustrate this well. Third, it gives a compact way to model the distribution changes. In realistic situations, data distribution might change over time. Fortunately, given the high-dimensional input, the changes often occur in a relatively small space in a causally-factorized system, which is known as the *minimal change principle* (Ghassami et al., 2018; Huang et al., 2020)

---

[1]Here, the definition of "non-parametric" is not about the general form of mapping function but indicates the functional causal model which takes the cause variables and errors as the input of a general function. Unlike the additive noise form, there is no constraint for how the noise interacts with the cause variable. Formal definition can be found in line 4 below Eq. (1.40) in (Pearl, 2009)

or *sparse mechanism shift* (Schölkopf et al., 2021). We can thus capture the distribution changes with low-dimensional change factors in a causal system instead of in the high-dimensional input space.

In summary, our main contributions are as follows:

- We propose a general formulation of SSMs, namely, NPSSM, together with its extension to allow nonstationarity of the latent process over time, which provides a flexible form for the transition and emission model that is expected to be widely applicable;
- We establish the identifiability of the time-lagged latent variables and their influencing strengths for NPSSM under relatively mild conditions;
- Based on our identifiability analysis, we propose a new structural VAE for model estimation and use it for forecasting tasks;
- Estimation of the proposed model can be seen as a way to learn the underlying temporal causal processes, which further facilitates forecasting of the time series;
- We evaluate the proposed method on a number of synthetic and real-world datasets. Experimental results demonstrate that latent causal dynamics could be reliably identified from observed data under various settings and further verify that identifying and using the latent temporal causal processes consistently improves the prediction performance.

## 2 PROBLEM FORMULATION

### 2.1 NPSSM: NON-PARAMETRIC STATE-SPACE MODEL AND IDENTIFIABILITY

To make SSMs in Eq. (1) flexible, we adopt the functional causal model (Pearl, 2009) to characterize transition process. Specifically, each latent factor $z_{it}$ is represented with a general form of structural causal model $z_{it} = f_i(\{z_{j,t-\tau} | z_{j,t-\tau} \in \text{Pa}(z_{it})\}, \epsilon_{it})$, where $i, j$ denotes variable element index, $\text{Pa}(z_{it})\}$ (parents) denotes the set of time-lagged variables that directly determine the latent factor $z_{it}$, and $\tau$ denotes the time lag index. In this way, noise $\epsilon_{it}$ together with parents of $z_{it}$ generate $z_{it}$ via unknown function $f(\cdot)$. Formally, NPSSM can be formulated as

$$\underbrace{z_{it} = f_i(\{z_{j,t-\tau} | z_{j,t-\tau} \in \text{Pa}(z_{it})\}, \epsilon_{it}),}_{\text{Structural causal latent transition}} \quad \underbrace{\mathbf{x}_t = g(\mathbf{z}_t, \eta_t) = g_1(g_2(\mathbf{z}_t) + \eta_t),}_{\text{Post nonlinear emission}} \quad (2)$$

where $\epsilon_{it}$ are mutually independent (i.e. spatially and temporally independent) random noises sampled from noise distribution $p(\epsilon_{it})$. $g_1(\cdot)$ is the invertible post-nonlinear distortion function, $g_2(\cdot)$ is the nonlinear mixing function and $\eta_t$ are independent noises (detailed notations are given in Appendix A2.1). To the best of our knowledge, this is the most general form of SSMs. In this transition function, the effect $z_{it}$ is just a smooth function (it refers to condition 3 of Theorem 1, which is the core condition to guarantee the identifiability of NPSSM) of its parents $\text{Pa}(z_{it})$ and noise $\epsilon_{it}$, and it contains linear models, nonlinear models with additive noise, and even multiplicative noise models as special cases. The Independent Noise condition and Conditional Independent condition (Pearl, 2009) are widely satisfied in time series data. Furthermore, in the emission function, the post-nonlinear transformation $g_1(\cdot)$ can model sensor or measurement distortion that usually happens when the underlying processes are measured with instruments (Zhang & Hyvarinen, 2012; Zhang & Hyvärinen, 2010).

Now, we define the identifiability of NPSSM in the function space. Once the latent variables $\mathbf{z}_1, \ldots, \mathbf{z}_T$ are identifiable up to componentwise transformations and permutation, latent transition (causal relationships) are also identifiable because conditional independence relations fully characterize time delayed causal relations in a time-delayed causally sufficient system. Therefore, we can say that NPSSM is identifiable if the latent variables are identifiable.

**Definition 1** (Identifiability of NPSSM). *For a ground truth $(f, g, p(\epsilon))$ and a learned $(\hat{f}, \hat{g}, \hat{p}(\epsilon))$ models as defined in Eq. (2), if the joint distribution for observed variables $p_{f,g,p(\epsilon)}(\mathbf{x}_t)$ and $p_{\hat{f},\hat{g},\hat{p}(\epsilon)}(\mathbf{x}_t)$ are matched almost everywhere, then we can say NPSSM are identifiable if observational equivalence can always lead to identifiability of the latent variables up to permutation $\pi$ and component-wise invertible transformation $T$:*

$$p_{\hat{g},\hat{f},\hat{p}_\epsilon}(\mathbf{x}_t) = p_{g,f,p_\epsilon}(\mathbf{x}_t) \Rightarrow g^{-1} = \hat{g}^{-1} \circ T \circ \pi. \quad (3)$$

*where $g^{-1}, \hat{g}^{-1}$ are invertible functions that maps $\mathbf{x}_t$ to $\mathbf{z}_t$ and $\hat{\mathbf{z}}_t$, respectively.*

Here we present the identifiability result of the proposed model. W.l.o.g., we assume the maximum time lag $L = 1$ in our analysis. Note that it is trivial to extend our analysis for long lag $L > 1$. We can see that, somewhat surprisingly, although NPSSM is remarkably flexible, it is actually identifiable up to relative minimum indeterminacies. Each latent process can be recovered up to its component-wise invertible transformations. In many real-world time series applications, these indeterminacies may be inconsequential.

**Theorem 1.** *Suppose that we observe data sampled from a generative model defined according to 2 with parameters $(\hat{f}, \hat{g}, \hat{p}(\epsilon))$. Assume the following holds:*

1. *The set $\{\mathbf{x}_t \in \mathcal{X} | \varphi_{\eta_t}(\mathbf{x}_t) = 0\}$ has measure zero, where $\varphi_{\eta_t}$ is the characteristic function of the density $p(\eta_t) = p_g(\mathbf{x}_t | \mathbf{z}_t)$. The post nonlinear functions $g_1, \hat{g}_1$ are invertible. The mixing functions $g_2, \hat{g}_2$ are injective and differentiable almost everywhere.*

2. *The process noise terms $\epsilon_{it}$ are mutually independent.*

3. *Let $\eta_{kt} \triangleq \log p(\mathbf{z}_{kt} | \mathbf{z}_{t-1})$, $\eta_{kt}$ is twice differentiable in $\mathbf{z}_{kt}$ and is differentiable in $\mathbf{z}_{l,t-1}, l = 1, 2, \ldots, n$. For each value of $\mathbf{z}_t, \mathbf{v}_{1t}, \mathring{\mathbf{v}}_{1t}, \mathbf{v}_{2t}, \mathring{\mathbf{v}}_{2t}, \ldots, \mathbf{v}_{nt}, \mathring{\mathbf{v}}_{nt}$ as $2n$ vector functions in $\mathbf{z}_{1,t-1}, \mathbf{z}_{2,t-1}, \ldots, \mathbf{z}_{n,t-1}$, are linearly independent, with $\mathbf{v}_{kt}$ and $\mathring{\mathbf{v}}_{kt}$ defined below:*

$$\mathbf{v}_{k,t} \triangleq \left( \frac{\partial^2 \eta_{kt}}{\partial z_{k,t} \partial z_{1,t-1}}, \frac{\partial^2 \eta_{kt}}{\partial z_{k,t} \partial z_{2,t-1}}, \ldots, \frac{\partial^2 \eta_{kt}}{\partial z_{k,t} \partial z_{n,t-1}} \right)^{\mathsf{T}}, \quad \mathring{\mathbf{v}}_{k,t} \triangleq \left( \frac{\partial^3 \eta_{kt}}{\partial z_{k,t}^2 \partial z_{1,t-1}}, \frac{\partial^3 \eta_{kt}}{\partial z_{k,t}^2 \partial z_{2,t-1}}, \ldots, \frac{\partial^3 \eta_{kt}}{\partial z_{k,t}^2 \partial z_{n,t-1}} \right)^{\mathsf{T}}.$$

*then $\mathbf{z}_t$ must be an invertible, component-wise transformation of a permuted version of $\hat{\mathbf{z}}_t$.*

The proofs are provided in Appendix A2.2. Theorem 1 indicates that we can find the underlying causal latent processes from the observed data. The differentiability and linear independence in condition 3 is the core condition to assure the identifiability of latent factors $\mathbf{z}_t$ from observed $\mathbf{x}_t$. It indicates that time-lagged variables must have a sufficiently complex and diverse effect on the transition distributions in terms of the second- and third-order partial derivatives. From this condition, we can find that the linear Gaussian SSM is unidentifiable since the second- and third-order partial derivatives would be constant, which violates the linear independence assumption.

## 2.2 N-NPSSM: Non-stationary Non-Parametric State Space Model

Considering that time series are non-stationary in many real situations, we now add a higher-level model to NPSSM to allow it to capture the time-varying characteristics of the process. We propose the Non-stationary Non-Parametric State Space Model(N-NPSSM), which is formulated as

$$\underbrace{\mathbf{x}_t = g_1(g_2(\mathbf{z}_t) + \eta_t)}_{\text{Post Nonlinear emission}}, \underbrace{\mathbf{z}_{it} = f_i(\{\mathbf{z}_{j,t-\tau} | \mathbf{z}_{j,t-\tau} \in \text{Pa}(\mathbf{z}_{it})\}, \mathbf{c}_t, \epsilon_{it})}_{\text{Structural causal latent transition}}, \underbrace{\mathbf{c}_t = f_c(\{\mathbf{c}_{t-\tau}\}_{\tau=1}^{L_c}, \zeta_t)}_{\text{Time-varying change factors}}, \quad (4)$$

where $\zeta_t$, similar to $\epsilon_{it}$, are mutually independent (i.e., spatially and temporally independent) random noises. $f_c(\cdot)$ is the transition function for the time-varying change factors, which is also formulated in a general form of a structural causal model. It includes the vanilla SSMs in Eq. (1) as a particular case in which the time-varying change factors do not exist. It also includes the time-varying parameter vector autoregressive model (Lubik & Matthes, 2015) as a special case, which allows the coefficients or the variances of noises in the linear auto-regressive model to vary over time following a specified law of motion. In contrast to explicitly specifying how time-varying change factors affect the transition process, our model is quite general in that we use a low-dimensional vector $\mathbf{c}_t$ to characterize the time-varying information and use it as an input for the transition model. Establishing the theoretical identifiability of this model is technically even more challenging, and our empirical results on various simulated data sets strongly suggest that it is actually identifiable.

## 3 Estimation Framework

Given our identifiability results, we propose the estimation procedures of NPSSM in Eq. (2) and N-NPSSM in Eq. 4). Since NPSSM is a special case of N-NPSSM, below, we consider only the estimation framework of N-NPSSM, and its properly constrained version will apply to NPSSM. The model architecture is shown in Fig. 2(a). Here $\mathbf{x}_t$ and $\hat{\mathbf{x}}_t$ are the observed and reconstructed variables. Similarly, $\mathbf{z}_t$ and $\hat{\mathbf{z}}_t$ denote the truth and estimated latent variables. The overall framework is a structural variational auto-encoder which learns the underlying latent temporal process via the **latent causal model** and then build the **auxiliary latent prediction model** on the uncovered latent variables. The implementation details are in Appendix A4.

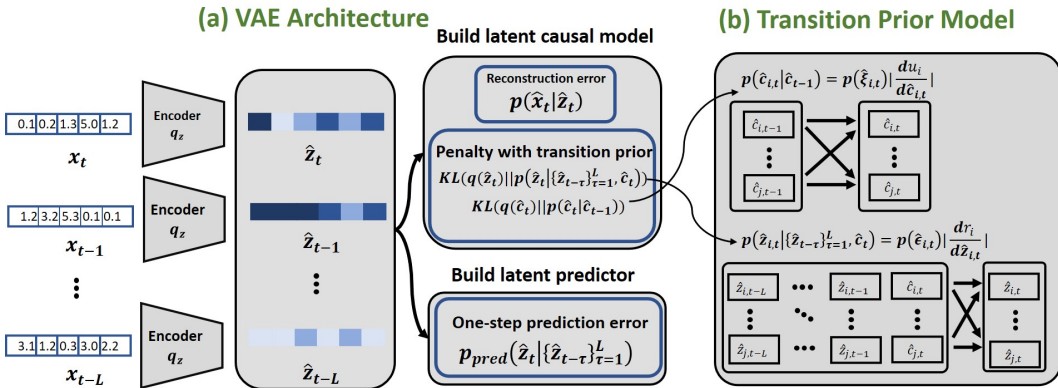

Figure 2: Fig (a) demonstrates the overview of our structural VAE framework. It mainly includes the latent causal model and latent prediction model. In latent causal model, it recovers latent process via minimizing reconstruction error and the regularization between factorized posterior $q(\hat{\mathbf{z}}_{1:T})$, $q(\hat{\mathbf{c}}_{1:T})$ and transition prior $p(\hat{\mathbf{z}}_{1:T})$, $p(\hat{\mathbf{c}}_{1:T})$, which implicitly models the temporal dynamics. Fig (b) shows the transition prior model, representing the latent causal processes $\hat{\mathbf{z}}_t$ and $\hat{\mathbf{c}}_t$.

**Latent Causal Model**   To facilitate our implementation, we adopt the Variational Auto-Encoder (Hsu et al., 2017), which implicitly implies that the measurement noise is additive. This is a particular case of the post-nonlinear mixing procedure given in Eq. (2). It is challenging to model the causal dependencies among observed and latent variables, especially for the design of the encoder/decoder. An alternative is to follow dynamic VAE (Girin et al., 2020) to encode the latent causal relationships in the encoder explicitly. To make the estimation more efficient, inspired by (Klindt et al., 2020; Yao et al., 2022), we use the transition prior $p(\hat{\mathbf{z}}_1, \ldots, \hat{\mathbf{z}}_T) = p(\hat{\mathbf{z}}_1) \cdots p(\hat{\mathbf{z}}_L) \prod_{t=L+1}^{T} p(\hat{\mathbf{z}}_t | \{\hat{\mathbf{z}}_{t-\tau}\}_{\tau=1}^{L}, \hat{\mathbf{c}}_t)$ and $p(\hat{\mathbf{c}}_1, \ldots, \hat{\mathbf{c}}_T) = p(\hat{\mathbf{c}}_1) \prod_{t=2}^{T} p(\hat{\mathbf{c}}_t | \hat{\mathbf{c}}_{t-1})$ to encode latent causal relationships and approximate the joint probability of posterior on $\mathbf{z}_{1:T}$ and $\mathbf{c}_{1:T}$ with factorized form. Specifically, the posterior(encoder) for $\mathbf{z}_{1:T}$ is defined as $\prod_{t=1}^{T} q(\hat{\mathbf{z}}_t | \mathbf{x}_t)$, and similarly the posterior(encoder) for $\mathbf{c}_{1:T}$ is defined as $\prod_{t=1}^{T} q_c(\hat{\mathbf{c}}_t | \{\hat{\mathbf{z}}_{t-\tau}\}_{\tau=0}^{L})$.

An alternative to model the transition prior probability (**transition prior model**) $p(\hat{\mathbf{z}}_t | \{\hat{\mathbf{z}}_{t-\tau}\}_{\tau=1}^{L}, \hat{\mathbf{c}}_t)$ is to leverage the forward prediction function $\hat{\mathbf{z}}_{it} = f_i(\{\hat{\mathbf{z}}_{j,t-\tau}\}_{\tau=1}^{L}, \hat{\mathbf{c}}_t, \hat{\epsilon}_{it})$. However, we argue that forward prediction with fixed loss cannot model latent processes without parametric form. For example, the latent process $z_{k,t} = q_k(\{\mathbf{z}_{t-\tau}\}_{\tau=1}^{L}) + \frac{1}{b_k(\{\mathbf{z}_{t-\tau}\}_{\tau=1}^{L})} \epsilon_{k,t}$ cannot be estimated by forward prediction function with squared loss because of the coupling effect from noise variable and cause variable. Thus, we propose to obtain this transition prior by explictly model the noise function, which can be treated as inverse latent transition function, i.e. $\hat{\epsilon}_{it} = r_i(\hat{z}_{it}, \hat{\mathbf{c}}_t, \{\hat{\mathbf{z}}_{t-\tau}\}_{\tau=1}^{L})$. Particularly, they are implemented by a set of separate MLP Networks $\{r_i\}$ (to satisfy the independent noise condition in Thm 1), which take the estimated latent causal variables and time-varying change factors as input and output the noise terms. By applying the change of variables formula to the transformation, the transition probability can be formulated as:

$$p\left(\hat{z}_{it} | \{\hat{\mathbf{z}}_{t-\tau}\}_{\tau=1}^{L}, \hat{\mathbf{c}}_t\right) = p_{\epsilon_{it}}\left(r_i(\hat{z}_{it}, \hat{\mathbf{c}}_t, \{\hat{\mathbf{z}}_{t-\tau}\}_{\tau=1}^{L})\right) \left| \frac{\partial r_i}{\partial \hat{z}_{it}} \right|. \tag{5}$$

Because of the mutually independent noise assumption, the Jacobian is a lower-triangular. We can efficiently calculate its determinant as the product of each element. By applying the independent noise assumption, the transition probability can be formulated as:

$$\log p(\hat{\mathbf{z}}_t | \{\hat{\mathbf{z}}_{t-\tau}\}_{\tau=1}^{L}, \hat{\mathbf{c}}_t) = \sum_{i=1}^{n} \log p(\hat{\epsilon}_{it}) + \sum_{i=1}^{n} \log \left| \frac{\partial r_i}{\partial \hat{z}_{it}} \right|. \tag{6}$$

Given this, the transition probability $p(\hat{\mathbf{z}}_t | \{\hat{\mathbf{z}}_{t-\tau}\}_{\tau=1}^{L}, \hat{\mathbf{c}}_t)$ can be efficiently evaluated using the factorized noise distribution $\sum_{i=1}^{n} \log p(\hat{\epsilon}_{it})$. To fit the estimated noises terms, we model each noise distribution $p(\hat{\epsilon}_{it})$ as a transformation from the standard normal noise $\mathcal{N}(0,1)$ through function $s(\cdot)$, which can be formulated as $p(\hat{\epsilon}_{it}) = p_{\mathcal{N}(0,1)}\left(s^{-1}(\hat{\epsilon}_{it})\right) \left| \frac{\partial s^{-1}(\hat{\epsilon}_{it})}{\partial \hat{\epsilon}_{it}} \right|$. Fortunately, we do not

need to explicitly estimate the term $\left|\frac{\partial s^{-1}(\hat{\epsilon}_{it})}{\partial \hat{\epsilon}_{it}}\right|$, since inverse causal transition functions $\{r_i\}$ could compensate it. Similarly, we define the transition probability of change factors $\mathbf{c}_t$ as $\log p_c(\hat{\mathbf{c}}_t|\hat{\mathbf{c}}_{t-1}) = \sum_{i=1}^{n} \log p(\hat{\zeta}_{it}) + \sum_{i=1}^{n} \log \left|\frac{\partial u_i}{\partial \hat{\mathbf{c}}_{it}}\right|$, where $u_i$ denotes the inverse change transition function.

**Auxiliary Latent Prediction Model** While the above latent causal model could estimate latent variable $\hat{\mathbf{z}}_t$ in the non-parametric form, it could not explicitly model the forward prediction relationship which is required for the forecasting task. Therefore, we propose to train the auxiliary latent prediction models. With penalization hyperparameter, one can view this module as a regularization to enforce the temporal predictability of the learned latent processes, for the purpose of time series forecasting. Formally, the auxiliary latent prediction models is defined as $p_{pred}(\hat{\mathbf{z}}_t|\{\hat{\mathbf{z}}_{t-\tau}\}_{\tau=1}^{L}, \hat{\mathbf{c}}_t, \hat{\epsilon}_t)$, which takes the recovered latent variables $\{\hat{\mathbf{z}}_t\}_{t=1}^{T}$, change factor $\hat{\mathbf{c}}_t$ and noise $\hat{\epsilon}_t$ as the input. Note that $\hat{\mathbf{c}}_t$ is not available at time $t-1$ in prediction mode. One straightforward solution is to build an extra prediction model for change factor $\hat{\mathbf{c}}_t$. Interestingly, we can skip this step, since change factor $\mathbf{c}_t$ had to be inferred from the latent variables $\{\hat{\mathbf{z}}_{t-\tau}\}_{\tau=0}^{L}$ as well, like the definition of posterior(encoder) $q_c(\hat{\mathbf{c}}_t|\{\hat{\mathbf{z}}_{t-\tau}\}_{\tau=0}^{L})$. Therefore, we can directly learn the auxiliary latent predictor via $p_{pred}(\hat{\mathbf{z}}_t|\{\hat{\mathbf{z}}_{t-\tau}\}_{\tau=1}^{L}, \hat{\epsilon}_t)$. Specifically, we use the LSTM network to implement this predictor. The noise $\hat{\epsilon}_t$ is generated from the inverse latent transition function $r_i(\hat{z}_{it}, \hat{\mathbf{c}}_t, \{\hat{\mathbf{z}}_{t-\tau}\}_{\tau=1}^{L})$ in the training phase, while it is sampled from the standard normal distribution $\mathcal{N}(0,1)$ in the forecasting phase.

This way, the prediction procedure decouples the forecasting task into three steps: (1). The encoder recovers the latent causal representation from the observed data; (2). Next-step prediction is generated via the latent prediction model in the latent space; (3) prediction results are transformed into observation space by the decoder.

**Optimization** By taking into account the above two components, we jointly train the latent causal model and the latent prediction model with the following objective $\mathcal{L}$:

$$\mathcal{L} = \underbrace{\frac{1}{T}\sum_{t=1}^{T}\log p_z(\mathbf{x}_t|\mathbf{z}_t) - \beta D_{KL}(q_z(\hat{\mathbf{z}}_{1:T}|\hat{\mathbf{x}}_{1:T})|p(\hat{\mathbf{z}}_{1:T})) - \gamma D_{KL}(q_c(\hat{\mathbf{c}}_{1:T}|\hat{\mathbf{z}}_{1:T})|p(\hat{\mathbf{c}}_{1:T}))}_{\text{latent causal model}},$$

$$+ \underbrace{\frac{\sigma}{T}\sum_{t=1}^{T}\log p_{pred}(\hat{\mathbf{z}}_t|\hat{\epsilon}_t, \{\hat{\mathbf{z}}_{t-\tau}\}_{\tau=1}^{L})}_{\text{auxiliary latent predictor}}, \quad (7)$$

where $p_z(\mathbf{x}_t|\mathbf{z}_t)$ and $p_{pred}(\hat{\mathbf{z}}_t|\hat{\epsilon}_t, \{\hat{\mathbf{z}}_{t-\tau}\}_{\tau=1}^{L})$ denote the decoder distribution and prediction distribution, in which we use MSE loss for the likelihood.

## 4 RELATED WORK

**Identifiability of State-Space Models** It is well-known that the linear state space model with additive Gaussian noise is unidentifiable (Arun & Kung, 1990), thus can not recover the latent process. Under specific structural constraints on the transition matrix, (Xu, 2002) find it identifiable. (Zhang & Hyvärinen, 2011) further consider the linear non-Gaussian setting and prove that when the emission matrix is of full column rank and the transition matrix is of full rank, the model is fully identifiable. In the non-stationary environment, (Huang et al., 2019) prove that the time-varying linear causal model is identifiable if the additive noise is a stationary zero-mean white noise process. For the vector autoregressive model with the latent process, (Jalali & Sanghavi, 2011) show that if the interactions between observed variables are sparse, interactions between latent variables and observed variables are sufficient, the transition matrix can be identified. (Geiger et al., 2015) find that if the additional genericity assumptions hold and the exogenous noises are independent non-Gaussian, then the transition matrix is uniquely identifiable. In contrast, our work considers a remarkably flexible state space model, which does not require constraints like linear transition or additive noise. Even so, we find that the latent process is generally identifiable.

**Deep State-Space Models** To leverage advances in deep learning, (Chung et al., 2015; Fraccaro et al., 2016; Karl et al., 2016; Krishnan et al., 2017) draw connections between the state space models

and RNN and propose the dynamic VAE framework to model temporal data. For (Chung et al., 2015), they associate the latent variables in the state space model with the deterministic hidden states of RNN. As such, the transition model is nonlinearly determined by the RNN and the observation model. These works propose different variants of deep learning architectures to parameterize transition and emission models to enhance expressiveness. These models vary in how they define the generative and inference model and how they combine the latent dynamic variables with RNN to model temporal dependencies (Girin et al., 2020). Meanwhile, the training paradigm of these works is similar to the VAE methodology. Inference networks define a variational approximation to the intractable posterior distribution of the latent variables. Approximation inference is applied, which may lead to sub-optimal performance. To address it, (Fraccaro et al., 2017; Rangapuram et al., 2018; Becker et al., 2019) take advantage of Kalman filters/smoothers to estimate the exact posterior distribution. For (Fraccaro et al., 2017), they use the standard Gaussian linear dynamical system to model the latent temporal process. The hidden states of RNN are used to predict the parameters of this dynamical system to enable closed-form Bayesian inference. However, these methods require expensive matrix inversion operation and the linear transition model limits the expressiveness. An alternative (Zheng et al., 2017) is to use variational sequential Monte Carlo to draw samples from the posterior directly. Recently, (Klushyn et al., 2021) propose a constraint optimization framework to obtain accurate predictions of the dynamical system. They achieve it by combining amortized variational inference with classic Bayesian filtering/smoothing to model dynamics. These works present different methods to infer the latent variables more accurately. Besides, some work leverage neural SDE to model the transition process (Yildiz et al., 2019). While these works enhance the expressivity of the transition model with deep architectures, they are still constrained by the additive noise form, which can be treated as special cases of our work.

## 5 EXPERIMENTS

To show the efficacy of N-NPSSM for identifying latent processes and forecasting, we apply it to various synthetic and real-world datasets with one-step-ahead forecasting tasks.

**Evaluation Metrics** To evaluate the identifiability of the learned latent variables, we report Mean Correlation Coefficient (MCC), which is a standard metric in ICA literature for continuous variables. We use Spearman's rank correlation coefficients to measure the discrepency between the ground-truth and estimated latent factors after component-wise transformation and permutation are adjusted (details are given in Appendix A3.2). MCC reaches 1 when latent variables are identifiable up to componentwise invertible transformation and permutation. To evaluate the forecasting performance, we report the Mean Absolute Error (MAE) and $\rho$-risk, which quantifies the accuracy of a quantile $\rho$ of the predictive distribution. Formally, they are defined as:

$$\text{MAE} = \sum_{i,t} |\mathrm{x}_{it} - \hat{\mathrm{x}}_{it}|, \quad R_\rho\text{-loss} = \sum_{i,t} (\hat{\mathrm{x}}_{it}^\rho - \mathrm{x}_{it})(\rho \mathbf{I}_{\hat{\mathrm{x}}_{it}^\rho > \mathrm{x}_{it}} - (1-\rho)\mathbf{I}_{\hat{\mathrm{x}}_{it}^\rho \leq \mathrm{x}_{it}}), \quad (8)$$

where $\hat{\mathrm{x}}_{it}^\rho$ is the empirical $\rho$-quantile of the prediction distribution and $\mathbf{I}$ is the indicator function. For the probabilistic forecasting models, forecast distribution is estimated by 50 trials of sampling, and $\hat{\mathrm{x}}_{it}$ is calculated by the predicted median value.

**Baselines** We compare N-NPSSM with typical deep forecasting models and deep state-space models: **(1)** LSTM(Hochreiter & Schmidhuber, 1997) which is a baseline for the deterministic deep forecasting model; **(2)** DeepAR(Salinas et al., 2020) which is an encoder-based probabilistic deep forecasting model; **(3)** VRNN(Chung et al., 2015) and **(4)** KVAE(Fraccaro et al., 2017) which are deep state space models. Note that KVAE implicitly considers time-varying change factors by formulating the transition matrix as a weighted average of a set of base matrices and using an RNN to predict the combination weights at each step.

### 5.1 SYNTHETIC EXPERIMENTS

We generate synthetic datasets that satisfy the identifiability conditions in the theorems. In particular, we consider four representative simulation settings to validate the identifiability and forecasting performance under fixed causal dynamics (Synthetic1), fixed causal dynamics with distribution shift (Synthetic2), time-varying causal dynamics with inter-dependent change factors (Synthetic3) and time-varying causal dynamics with changing causal strengths (Synthetic4) (more details of data generation are given in Appendix A3.1.1). For all the synthetic datasets, we set latent size $n = 8$, and

Table 1: Identifiability and forecasting performance for the four synthetic datasets (more empirical results can be found in A3.3). Note: "N/A" indicates the deterministic model LSTM is not applicable to output predictive distribution

| Method | Synthetic 1 | | | Synthetic 3 | | |
|---|---|---|---|---|---|---|
| | MCC | MAE | $R_{0.9}$-loss | MCC | MAE | $R_{0.9}$-loss |
| LSTM | 0.110±0.02 | 0.416±0.03 | N/A | 0.140±0.02 | 0.583±0.03 | N/A |
| KVAE | 0.406±0.02 | 0.404±0.01 | 2.237±0.05 | 0.513±0.05 | 0.455±0.02 | 6.446±0.04 |
| VRNN | 0.520±0.08 | 0.515±0.02 | 4.341±0.06 | 0.555±0.03 | 0.543±0.01 | 3.578±0.03 |
| DeepAR | 0.267±0.03 | 0.087±0.03 | 0.353±0.03 | 0.432±0.02 | 0.095±0.02 | 0.606±0.02 |
| N-NPSSM | **0.987±0.01** | **0.054±0.01** | **0.220±0.03** | **0.998±0.01** | **0.057±0.01** | **0.363±0.01** |

| Method | Synthetic 2 | | | Synthetic 4 | | |
|---|---|---|---|---|---|---|
| | MCC | MAE | $R_{0.9}$-loss | MCC | MAE | $R_{0.9}$-loss |
| LSTM | 0.199±0.02 | 0.498±0.08 | N/A | 0.227±0.02 | 0.641±0.02 | N/A |
| KVAE | 0.407±0.02 | 0.479±0.05 | 25.94±0.61 | 0.478±0.03 | 0.480±0.01 | 2.090±0.02 |
| VRNN | 0.491±0.08 | 0.637±0.06 | 28.58±0.83 | 0.397±0.03 | 0.498±0.02 | 1.167±0.01 |
| DeepAR | 0.297±0.01 | 0.133±0.03 | 3.284±0.04 | 0.351±0.03 | 0.087±0.01 | 0.179±0.01 |
| N-NPSSM | **0.995±0.01** | **0.069±0.01** | **1.866±0.02** | **0.992±0.01** | **0.081±0.01** | **0.169±0.02** |

the maximum latent process lag is set to $L = 2$. For time-varying settings, the dimension of change variables is set to 4. The emission function $g(\cdot)$ is a random three-layer MLP with LeakyReLU units.

As shown in Table 1, N-NPSSM can successfully recover the latent processes under different settings, as indicated by the highest MCC (close to 1). In contrast, the baseline models, including the deep forecasting model and deep state-space models, cannot recover the latent processes. Besides, our method gives the best forecasting accuracy, as indicated by the lowest MAE and $R_{0.9}$-loss. In Figure 4, each left sub-figure shows the MCC correlation matrix of each factor, while each right sub-figure shows the scatter plot of recovered factors and truth factors. We can find that the time-delayed causal relationships are successfully recovered, as indicated by high MCC for the causally-related factors. Besides, the latent causal variables are estimated up to permutation and componentwise invertible transformation (more empirical results are given in A3.3).

To investigate the consequence of the violation of the critical assumptions. We create another two datasets: (1) with dependent process noise terms, and (2) with additive Gaussian noise terms, in which (1) violates the mutually independent noise condition, and (2) violates the linear independence condition. From Figure 3, we can find that violating the independent noise condition deteriorates the identifiability results significantly. Additionally, when the latent processes follow a linear, additive Gaussian temporal model (thus, the linear independence condition is violated), the identifiability results are also distorted. However, if the noise terms are slightly non-Gaussian (we change the shape parameter $\beta$ of the generalized Gaussian noise distribution from $\beta = 2.0$ to $\beta = 1.5$ or $\beta = 2.5$, we can observe the final MCC scores increase significantly and the underlying latent processes become identifiable in both non-Gaussian noise scenarios.

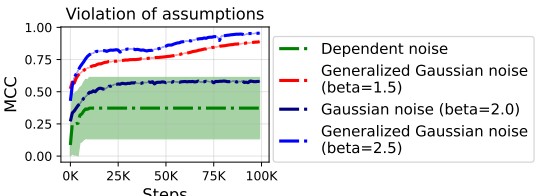

Figure 3: MCC trajectories of NPSSM for temporal data with clear assumption violations.

## 5.2 REAL DATA EXPERIMENTS

We evaluate N-NPSSM on three real-world datasets: Economics, Bitcoin and FRED. Economics and Fred contain a set of macroeconomic indicators, while Bitcoin includes the potential influencers of the bitcoin price (The detailed data descriptions and preprocess are given in Appendix A3.1.2). As shown in Table 2, N-NPSSM outperforms all competitors in terms of both MAE and $R_{0.9}$-loss, which verifies the effectiveness of N-NPSSM (more qualitative experiments are given in Appendix A3.3).

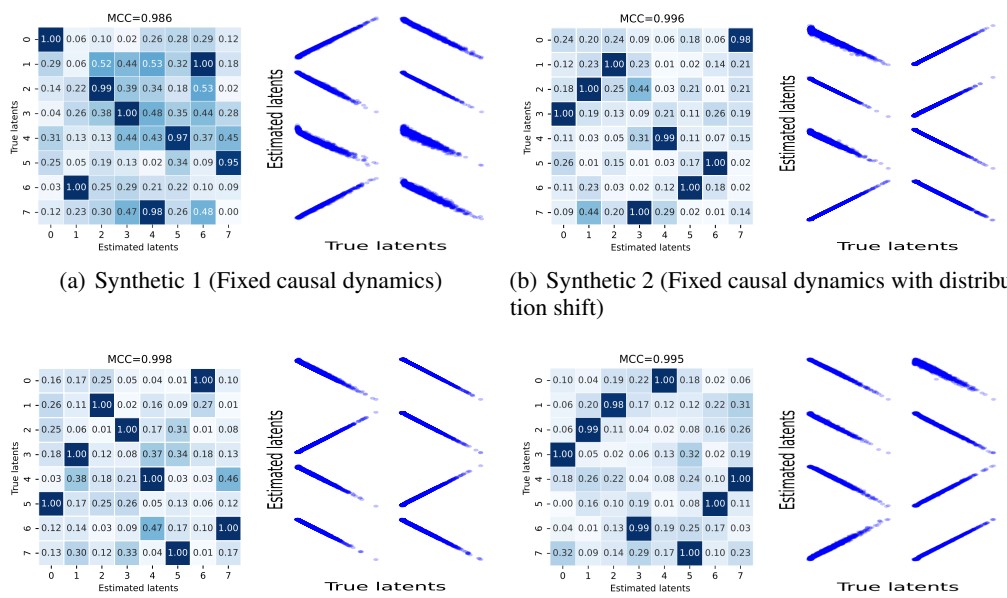

(a) Synthetic 1 (Fixed causal dynamics)

(b) Synthetic 2 (Fixed causal dynamics with distribution shift)

(c) Synthetic 3 (Time-varying causal dynamics with inter-dependent change factors)

(d) Synthetic 4 (Time-varying causal dynamics with changing causal strength)

Figure 4: MCC for causally-related factors and scatter plots between estimated factors and true factors on four synthetic datasets.

Table 2: Forecasting performance on three real-world datasets

| Method | Economics | | Bitcoin | | FRED | |
|---|---|---|---|---|---|---|
| | MAE | $R_{0.9}$-loss | MAE | $R_{0.9}$-loss | MAE | $R_{0.9}$-loss |
| LSTM | 0.717±0.04 | N/A | 0.747±0.04 | N/A | 0.632±0.05 | N/A |
| KVAE | 0.618±0.01 | 1.363±0.10 | 0.551±0.01 | 0.290±0.03 | 0.619±0.03 | 0.883±0.04 |
| VRNN | 0.786±0.12 | 1.534±0.07 | 0.759±0.06 | 0.222±0.01 | 0.728±0.01 | 1.045±0.08 |
| DeepAR | 0.741±0.08 | 1.288±0.12 | 1.465±0.01 | 0.317±0.05 | 0.752±0.05 | 0.654±0.04 |
| N-NPSSM | **0.603±0.05** | **1.190±0.11** | **0.403±0.01** | **0.143±0.01** | **0.484±0.03** | **0.580±0.05** |

## 6 CONCLUSION AND FUTURE WORK

In this work, we propose a general formulation of state-space models called NPSSM, which includes a completely non-parametric transition model and a flexible emission model. We prove that even though it is flexible, it is generally identifiable. Moreover, we further propose N-NPSSM to capture the possible time-varying change property of the latent processes. We further develop the estimation procedure based on VAE and make use of it for forecasting tasks. Empirical studies on both synthetic and real-world datasets validate that our model could not only identify the latent process but also outperform baselines in the forecasting task. While we do not establish theories with time-varying change factors, we have demonstrated through experiments the possibilities of generalizing our identifiability results to this setting. Extending our theories to address the issue of a completely non-parametric emission model will also be one line of our future work. Another interesting direction is to apply this framework to other time series analysis intelligence tasks, like anomaly detection and change point detection, which is also interesting directions.

## REPRODUCIBILITY STATEMENT

Our code for NPSSM is attached as supplementary material. The implementation details can be found in A4. For theoretical results, the assumptions and complete proof of the claims are in A2.2. For synthetic experiments, the data generation process is described in A3.1.1.

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

*Appendix for*

## "Non-Parametric State-Space Models: Identifiability, Estimation and Forecasting"

Appendix organization:

## A1    EXTENDED RELATED WORK

**Time-Varying State-Space Models**    In many real situations, the temporal process may vary over time. This inspired the early efforts to allow the parameters of vector autoregressive models to change over time (Sodsri, 2003; Luo, 2005), which consider the effect of time variation in coefficients and the variance of noises. These works can be treated as special cases of the state space models, which directly learn the transition in observation space. Time-varying linear state space models (Luttinen et al., 2014; Holmes et al., 2012) make one step further, as it is more powerful and general than vector autoregressive models. A similar research topic is the switching-regime state space models (Ghahramani & Hinton, 1996; 2000; Glaser et al., 2020), which assumes the transition lies in a set of linear dynamical models and model the transition process with hidden Markov models. Thus, these models cannot capture the continuous change over time. Recently, some deep state space models have implicitly considered the time-varying characteristic of data. Both of these works (Rangapuram et al., 2018; Fraccaro et al., 2017) consider the Gaussian linear dynamical systems in the latent space. In (Rangapuram et al., 2018), the transition/emission matrices and two noise covariance matrices are predicted by RNN at each step. In (Fraccaro et al., 2017), they assume the transition/emission matrices are a weighted average of a set of base matrices, where the RNN model predicts the weights

at each step. Note that all these existing works require specifying how time-varying change factors affect the transition process, which may not be applicable in practice without prior knowledge. In contrast, our model is flexible since we consider a more general transition model, and the time-varying change factors are treated as the input for the transition process.

## A2 IDENTIFIABILITY THEORY

### A2.1 NOTATIONS

We summarize the notations used throughout the paper in Table A1.

Table A1: Notations.

| Index | |
|---|---|
| $t$ | Time index |
| $i, j, k$ | Variable element (channel) index |
| $\tau$ | Time lag index |
| $L$ | Maximum time lag for latent variable |
| $L_c$ | Maximum time lag for time-varying change factors |
| **Variable** | |
| $\mathbf{x}_t$ | Observation data |
| $\hat{\mathbf{x}}_t$ | Reconstructed observation |
| $\mathbf{z}_t$ | latent variable |
| $\hat{\mathbf{z}}_t$ | estimated latent variable |
| $\mathbf{c}_t$ | time-varying change variable |
| $\hat{\mathbf{c}}_t$ | estimated time-varying change variable |
| $\mathbf{Pa}(z_{it}), \{\mathbf{z}_{t-\tau}\}$ | Set of direct cause nodes/parents of node $z_{it}$ |
| $\eta_t$ | measurement noise term |
| $\epsilon_{it}$ | Process noise term |
| $\zeta_t$ | noise term for time-varying change factor |
| **Function and Hyperparameter** | |
| $p$ | Distribution function (e.g., $p_{\epsilon_{it}}$ is the distribution of $\epsilon_{it}$.) |
| $g$ | Arbitrary nonlinear and invertible mixing function |
| $f_i$ | Nonlinear transition function for $z_{it}$ |
| $f_c$ | Nonlinear transition function for $\mathbf{c}_t$ |
| $h$ | Post nonlinear distortion function |
| $r_i$ | Learned inverse transition function for residual $\hat{\epsilon}_i$ |
| $u_i$ | Learned inverse change transition function for residual $\hat{\zeta}_i$ |
| $\beta, \gamma, \sigma$ | Weights in the augmented ELBO objective |
| $n$ | Latent size |
| $\pi$ | Permutation operation |
| $T$ | Component-wise invertible nonlinearities |

### A2.2 PROOF OF IDENTIFIABILITY THEORY

Before the proof, we first produce Lemma 1, which presents the identifiability of latent variables in fixed latent dynamics. This result will be used in the proof of Theorem 1.

**Lemma 1.** *(Theorem 1 in (Yao et al., 2022)) The fixed latent causal dynamics takes on the following form:*

$$\mathbf{x}_t = g(\mathbf{z}_t) \quad \mathbf{z}_{it} = f_i(\{\mathbf{z}_{j,t-1} | \mathbf{z}_{j,t-1} \in \text{Pa}(\mathbf{z}_{it})\}, \epsilon_{it}). \tag{9}$$

*Let $\eta_{kt} \triangleq \log p(\mathbf{z}_{kt}|\mathbf{z}_{t-1})$, $\eta_k(t)$ is twice differentiable in $\mathbf{z}_{kt}$ and is differentiable in $\mathbf{z}_{l,t-1}, l = 1, 2, \ldots, n$. Suppose there exists an invertible function $\hat{\mathbf{g}}$ that maps $\mathbf{x}_t$ to $\hat{\mathbf{z}}_t$, i.e., $\hat{\mathbf{z}}_t = \hat{\mathbf{g}}(\mathbf{x}_t)$, such*

*that the components of $\hat{\mathbf{z}}_t$ are mutually independent conditional on $\hat{\mathbf{z}}_{t-1}$. Let*

$$\mathbf{v}_{k,t} \triangleq \Big( \frac{\partial^2 \eta_{kt}}{\partial z_{k,t} \partial z_{1,t-1}}, \frac{\partial^2 \eta_{kt}}{\partial z_{k,t} \partial z_{2,t-1}}, ..., \frac{\partial^2 \eta_{kt}}{\partial z_{k,t} \partial z_{n,t-1}} \Big)^\mathsf{T},$$

$$\mathring{\mathbf{v}}_{k,t} \triangleq \Big( \frac{\partial^3 \eta_{kt}}{\partial z_{k,t}^2 \partial z_{1,t-1}}, \frac{\partial^3 \eta_{kt}}{\partial z_{k,t}^2 \partial z_{2,t-1}}, ..., \frac{\partial^3 \eta_{kt}}{\partial z_{k,t}^2 \partial z_{n,t-1}} \Big)^\mathsf{T}.$$

*If for each value of $\mathbf{z}_t$, $\mathbf{v}_{1,t}, \mathring{\mathbf{v}}_{1,t}, \mathbf{v}_{2,t}, \mathring{\mathbf{v}}_{2,t}, ..., \mathbf{v}_{n,t}, \mathring{\mathbf{v}}_{n,t}$, as 2n vector functions in $z_{1,t-1}, z_{2,t-1}, ..., z_{n,t-1}$, are linearly independent, then $\mathbf{z}_t$ must be an invertible, component-wise transformation of a permuted version of $\hat{\mathbf{z}}_t$.*

Second, we consider the additive noise model, in which $g_1$ is the identity mapping. To identify the noise-free distribution $g(\mathbf{z}_t)$ from noisy data with assumption 1, we follow the idea of using convolution theorem to decouple measurement error (Khemakhem et al., 2020). Since the volume of a matrix $vol\,\mathbf{A}$ is defined as the product of the singular values of $\mathbf{A}$. We could obtain that $vol\,\mathbf{A} = |det\,\mathbf{A}|$ when $\mathbf{A}$ is invertible. We use $vol\,\mathbf{A}$ in the change of variables formula to replace the absolute determinant of the Jacobian (Ben-Israel, 1999). Suppose the joint distribution for observed variables $p_{f,g,p(\epsilon)}(\mathbf{x}_t|\mathbf{z}_{t-1})$ and $p_{\hat{f},\hat{g},\hat{p}(\epsilon)}(\mathbf{x}_t|\hat{\mathbf{z}}_{t-1})$ are matched almost everywhere. Then:

$$\int_{\mathcal{Z}} p_{f,p(\epsilon)}(\mathbf{z}_t|\mathbf{z}_{t-1})p_g(\mathbf{x}_t|\mathbf{z}_t)d\mathbf{z}_t = \int_{\mathcal{Z}} p_{\hat{f},\hat{p}(\epsilon)}(\mathbf{z}_t|\hat{\mathbf{z}}_{t-1})p_{\hat{g}}(\mathbf{x}_t|\mathbf{z}_t)d\mathbf{z}_t, \tag{10}$$

$$\int_{\mathcal{Z}} p_{f,p(\epsilon)}(\mathbf{z}_t|\mathbf{z}_{t-1})p_{\eta_t}(\mathbf{x}_t - g(\mathbf{z}_t))d\mathbf{z}_t = \int_{\mathcal{Z}} p_{\hat{f},\hat{p}(\epsilon)}(\mathbf{z}_t|\hat{\mathbf{z}}_{t-1})p_{\eta_t}(\mathbf{x}_t - \hat{g}(\mathbf{z}_t))d\mathbf{z}_t. \tag{11}$$

According to the Jacobian matrix of the mapping from $\bar{\mathbf{x}}_t = g(\mathbf{z}_t)$ and $\bar{\mathbf{x}}_t = \hat{g}(\mathbf{z}_t)$, we have

$$\int_{\mathcal{X}} p_{f,p(\epsilon)}(g^{-1}(\bar{\mathbf{x}}_t)|\mathbf{z}_{t-1})vol\,\mathbf{J}_{g^{-1}}(\bar{\mathbf{x}}_t)p_{\eta_t}(\mathbf{x}_t - \bar{\mathbf{x}}_t)d\bar{\mathbf{x}}_t$$
$$= \int_{\mathcal{X}} p_{\hat{f},\hat{p}(\epsilon)}(\hat{g}^{-1}(\bar{\mathbf{x}}_t)|\hat{\mathbf{z}}_{t-1})vol\,\mathbf{J}_{g^{-1}}(\bar{\mathbf{x}}_t)p_{\eta_t}(\mathbf{x}_t - \bar{\mathbf{x}}_t))d\bar{\mathbf{x}}_t. \tag{12}$$

Let us assume $\bar{p}_{f,p(\epsilon),g,\mathbf{z}_{t-1}}(\mathbf{x}_t) = p_{f,p(\epsilon)}(g^{-1}(\mathbf{x}_t)|\mathbf{z}_{t-1})vol\,\mathbf{J}_{g^{-1}}\mathbb{I}_{\mathcal{X}}(\mathbf{x}_t)$, and then we have

$$\int_{\mathcal{X}} \bar{p}_{f,p(\epsilon),g,\mathbf{z}_{t-1}}(\bar{\mathbf{x}}_t)p_{\eta_t}(\mathbf{x}_t - \bar{\mathbf{x}}_t)d\bar{\mathbf{x}}_t = \int_{\mathcal{X}} \bar{p}_{\hat{f},\hat{p}(\epsilon),\hat{g},\hat{\mathbf{z}}_{t-1}}(\bar{\mathbf{x}}_t)p_{\eta_t}(\mathbf{x}_t - \bar{\mathbf{x}}_t))d\bar{\mathbf{x}}_t. \tag{13}$$

According to the convolution theorem (Katznelson, 2004) that the convolution in one domain (e.g., time domain) equals point-wise multiplication in the other domain (e.g., frequency domain). We could obtain that,

$$(\bar{p}_{f,p(\epsilon),g,\mathbf{z}_{t-1}} \star p_{\eta_t})(\mathbf{x}_t) = (\bar{p}_{\hat{f},\hat{p}(\epsilon),\hat{g},\hat{\mathbf{z}}_{t-1}} \star p_{\eta_t})(\mathbf{x}_t), \tag{14}$$

$$F[\bar{p}_{f,p(\epsilon),g,\mathbf{z}_{t-1}}](\omega)\varphi_{\eta_t}(\omega) = F[\bar{p}_{\hat{f},\hat{p}(\epsilon),\hat{g},\hat{\mathbf{z}}_{t-1}}](\omega)\varphi_{\eta_t}(\omega), \tag{15}$$

where $\star$ denotes the convolution operator and $F[\cdot]$ denotes the Fourier transform. We can find $\varphi_{\eta_t} = F[p_{\eta_t}]$ by the definition of characteristic function in Assumption 1. Then, we can remove $\varphi_{\eta_t}(\omega)$ the term from both sides, as it is non-zero almost everywhere. We have,

$$F[\bar{p}_{f,p(\epsilon),g,\mathbf{z}_{t-1}}](\omega) = F[\bar{p}_{\hat{f},\hat{p}(\epsilon),\hat{g},\hat{\mathbf{z}}_{t-1}}](\omega), \tag{16}$$

$$\bar{p}_{f,p(\epsilon),g,\mathbf{z}_{t-1}}(\mathbf{x}_t) = \bar{p}_{\hat{f},\hat{p}(\epsilon),\hat{g},\hat{\mathbf{z}}_{t-1}}(\mathbf{x}_t). \tag{17}$$

Thus, we can conclude that if the distributions are the same with additive noise, the noise-free distributions are still the same. Combined with the results from Lemma 1 that the latent variables are identifiable up to permutation and component-wise invertible transformation.

Lastly, we consider the effect of post non-linear function $g_1(\cdot)$. Let us denote $\tilde{\mathbf{x}}_t = g_2(\mathbf{z}_t) + \eta_t$, then the learned post non-linear function $\mathbf{x}_t = \hat{g}_1(\tilde{\mathbf{x}}_t)$ can be written as $\mathbf{x}_t = (g_1 \circ (g_1)^{-1} \circ \hat{g}_1)(\tilde{\mathbf{x}}_t)$. We can further assume that $\hat{g}_1 = g_1 \circ ((g_1)^{-1} \circ \hat{g}_1) = g_1 \circ g_3$, in which $g_3$ represents the indeterminacy on the space of $\tilde{\mathbf{x}}_t$. Following the proof of Theorem 1 of (Klindt et al., 2020), we have that $g_3$ can only be a bijection if both $g_2, \hat{g}_1$ are injective functions. Thus, we can treat it as adding a component-wise invertible nonlinear function $g_3^{-1}$ on $\mathbf{x}_t$, which does not affect the identifiability of $\mathbf{z}_t$ up to permutation and component-wise invertible transformation. Therefore, NPSSM in 9 is identifiable.

## A3 EXPERIMENT DETAILS

### A3.1 DATASETS

#### A3.1.1 SYNTHETIC DATASET GENERATION

To evaluate the identifiability and forecasting capability of our model under different scenarios, we generate the synthetic data with 1) fixed causal dynamics; 2) fixed causal dynamics with distribution shift; 3) time-varying causal dynamics with changing noise variances, and 4) time-varying causal dynamics with changing causal strengths. We use the first $80\%$ data for training and the rest $20\%$ for evaluation.

**Stationary Causal Dynamics**   For the fixed causal dynamics, we generate 100,000 data points based on the following equation:

$$z_{k,t} = q_k(\{\mathbf{z}_{t-\tau}\}) + \frac{1}{b_k(\{\mathbf{z}_{t-\tau}\})}\epsilon_{k,t}. \tag{18}$$

Here, $\epsilon_{k,t}$ is the process noise, which are sampled from i.i.d. Gaussian distribution ($\sigma = 0.1$). $\epsilon_{1,t}, \epsilon_{2,t}, .., \epsilon_{n,t}$ are mutually independent and independent of $\mathbf{z}_{t-1}$. The process noise terms are coupled with the history information through multiplication with the average value of all the time-lagged latent variables. We set the latent size $n = 8$ and the lag number of the process $L = 2$. We apply a 2-layer MLP with LeakyReLU as the state transition function. The emission function is a random three-layer MLP with LeakyReLU units.

**Stationary Causal Dynamics with Distribution Shift**   We follow the same way as the setting of fixed causal dynamics and generate 80,000 data points for the training set. To simulate distribution shift in the test phase, we vary the values of the first layer of the MLP in the test set and generate 20,000 samples. The entries of the kernel matrix of the first layer are uniformly distributed between [-1,1].

**Time-Varying Causal Dynamics with Changing Causal Strengths**   For the time-varying causal dynamics with changing causal strengths, we generate 100,000 data points based on the following equation:

$$c_{k,t} = c_{k,t-1} + \zeta_{k,t}$$
$$z_{k,t} = q_k(\{\mathbf{z}_{t-\tau}\}, \mathbf{c}_t) + \frac{1}{b_k(\{\mathbf{z}_{t-\tau}\})}\epsilon_{k,t}, \tag{19}$$

where the noises $\zeta_{kt}$ are sampled from i.i.d. Laplace distribution ($\sigma = 1$). We take the change factor $\mathbf{c}_t$ as an input for the latent transition function for $\mathbf{z}_t$.

**Time-Varying Causal Dynamics with Inter-Dependent Change Factors**   For the time-varying causal dynamics with inter-dependent change factors, instead of considering the independent sources using temporal dependencies, here we consider the inter-dependence across the different variable index. Formally we generate 100,000 data points based on the following equation:

$$\mathbf{c}_t = \boldsymbol{C}\mathbf{c}_{t-1} + \zeta_{k,t}$$
$$z_{k,t} = q_k(\{\mathbf{z}_{t-\tau}\}) + \frac{1}{b_k(\{\mathbf{z}_{t-\tau}\}, \mathbf{c}_t)}\epsilon_{k,t}, \tag{20}$$

where $\boldsymbol{C}$ is the transition matrix for change factors. The noises $\zeta_{kt}$ are sampled from i.i.d. Laplace distribution ($\sigma = 1$). In the latent transition process for $\mathbf{z}_t$, noise terms are coupled with the history information and change factors through multiplication with the average value of all the time-lagged latent variables $\mathbf{z}_{t-\tau}$ and current time-varying change factor $\mathbf{c}_t$.

#### A3.1.2 REAL-WORLD DATASET

Three real-world datasets are used to evaluate the forecasting performance of the proposed model. We use the first $80\%$ data for training and the rest $20\%$ for evaluation.

Table A2: Comparison with additional baselines on real-world datasets.

| Methods | Economics | | Bitcoin | | FRED | |
|---|---|---|---|---|---|---|
| | MAE | $R_{0.9}$-Loss | MAE | $R_{0.9}$-Loss | MAE | $R_{0.9}$-Loss |
| DKF | 0.784 | 1.877 | 0.516 | 0.148 | 0.722 | 0.901 |
| RVAE | 0.798 | 1.775 | 0.524 | 0.183 | 0.666 | 1.128 |
| SRNN | 0.710 | 1.266 | 0.530 | 0.202 | 0.787 | 1.289 |
| DSAE | 0.736 | 1.594 | 0.493 | 0.227 | 0.650 | 1.061 |
| N-NPSSM | **0.603** | **1.190** | **0.403** | **0.143** | **0.484** | **0.580** |

**Economics**  The economics dataset was used in (Huang et al., 2019). We investigate the time-lagged causal relationships among 10 macroeconomic variables ranging from CPI, inflation to unemployment rate with monthly data from 1965 to 2017 in the USA[2]. The data are normalized by subtracting the mean and dividing them by the standard deviation.

**Bitcoin**  The bitcoin dataset was used in (Godahewa et al., 2021). We investigate the time-lagged causal relationships about 16 daily time series [3], which have potential influences on the bitcoin price. Specifically, it includes hash rate, block size, mining difficulty, public opinion, etc. The data are normalized by subtracting the mean and dividing them by the standard deviation.

**FRED**  The FRED dataset was used in (Godahewa et al., 2021). We investigate the time-lagged causal relationships about 107 monthly time series [4]. It contains a set of macroeconomic indicators from the Federal Reserve bank. The data are normalized by subtracting the mean and dividing them by the standard deviation.

## A3.2    Evaluation Metric

**Mean Correlation Coefficient (MCC)**  MCC is a standard metric for evaluating the recovery of latent factors in ICA literature. We first apply a nonlinear regression to the recovered factors, aiming to get rid of the component-wise transformation indeterminacy, for each possible pair of the estimated factor and the true one. Then, we calculate all pairs of correlation coefficients (the absolute values of the Spearman's rank correlation coefficients) between ground-truth latent factors and the estimated latent factors (after the component-wise transformation). We further solve a linear sum assignment problem to assign each latent component to the ground-truth component that best correlates with it, thus finding the correspondence between the estimated factors and the true ones in the latent space. A high MCC means one successfully recovered the true latent factors, up to invertible, component-wise transformation and permutation.

## A3.3    Additional Experimental Results

### A3.3.1    Additional Time Series Forecasting Results

We list additional comparison with more baselines from typical deep state space models, including DKF(Krishnan et al., 2015), SRNN(Fraccaro et al., 2016), RVAE(Leglaive et al., 2020), DSAE(Yingzhen & Mandt, 2018). As shown in Table A2, our method still consistently outperforms the additional baselines.

In Table A3, we evaluate the proposed model N-NPSSM and baselines on more real-world datasets NN5[5], OIKOLAB[6] and Pedestrain[7]. We can find that N-NPSSM still outperforms the baselines.

---

[2]Downloaded from https://www.theglobaleconomy.com/

[3]Downloaded from https://zenodo.org/record/5122101#.YzPm7exBz0o

[4]Downloaded from https://zenodo.org/record/4654833#.YzPo1exBz0o

[5]https://zenodo.org/record/4656117

[6]https://zenodo.org/record/5184708

[7]https://zenodo.org/record/4656626

Table A3: Comparison with baselines on additional real-world datasets.

| Methods | NN5 | | OIKOLAB | | Pedestrain | |
|---------|-----|-----|---------|-----|------------|-----|
| | MAE | $R_{0.9}$-Loss | MAE | $R_{0.9}$-Loss | MAE | $R_{0.9}$-Loss |
| LSTM | 0.890 | N/A | 0.422 | N/A | 0.739 | N/A |
| KVAE | 0.573 | 8.858 | 0.424 | 1.190 | 0.362 | 6.475 |
| VRNN | 0.869 | 9.259 | 0.842 | 1.310 | 0.769 | 9.794 |
| DeepAR | 0.567 | 6.405 | 0.191 | 0.325 | 0.387 | 4.573 |
| N-NPSSM | **0.560** | **5.969** | **0.152** | **0.323** | **0.293** | **4.191** |

Table A4: Comparison with baselines on multi-step (3-step) time series forecasting setting.

| Methods | Economics | | Bitcoin | | Fred | |
|---------|-----------|-----|---------|-----|------|-----|
| | MAE | $R_{0.9}$-Loss | MAE | $R_{0.9}$-Loss | MAE | $R_{0.9}$-Loss |
| LSTM | 0.724 | N/A | 0.634 | N/A | 0.694 | N/A |
| DeepAR | 0.895 | 1.665 | 1.483 | 0.658 | 0.922 | 0.677 |
| KVAE | 0.612 | 1.365 | 0.551 | 0.299 | 0.534 | 0.795 |
| VRNN | 0.797 | 1.592 | 0.685 | 0.248 | 0.728 | 1.009 |
| N-NPSSM | **0.588** | **1.177** | **0.430** | **0.166** | **0.595** | **0.602** |

In Table A4, we compare N-NPSSM with the baselines on multi-step (3-step) forecasting setting. We can find that our proposed model N-NPSSM consistently outperforms the baselines.

### A3.3.2 ABLATION STUDIES

In table A5, we show the performance of N-NPSSM and NPSSM on synthetic datasets. We can find that N-NPSSM achieves comparable performance with NPSSM on fixed causal dynamics settings, while N-NPSSM has a higher MCC score on time-varying causal dynamics settings.

In table A6, we compare the identifiability results of NPSSM with/without transition prior network. We can find this module is critical to the identifiability of NPSSM.

### A3.3.3 EMPIRICAL STUDIES FOR VIOLATION ASSUMPTIONS

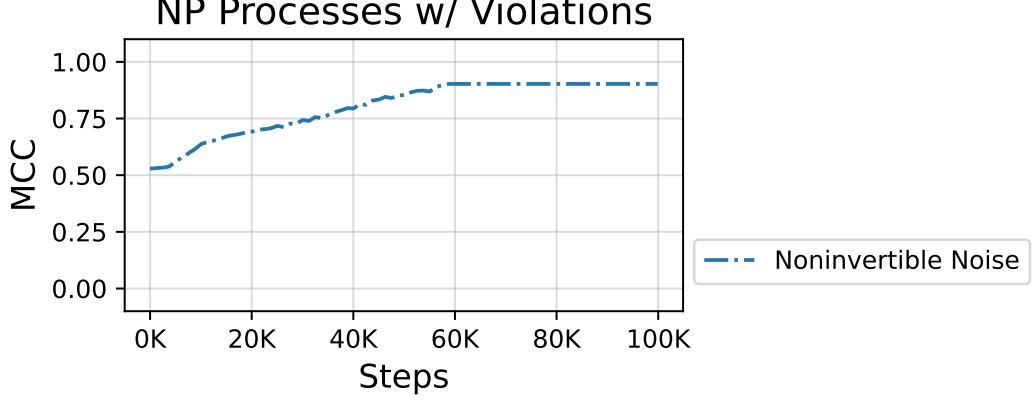

Figure A1: MCC trajectories of NPSSM for temporal data with non-inveritble noise.

To verify the effectiveness of NPSSM when noise function is non-inveritble, we consider the generation process of replacing the stationary causal dynamics in Eq. (eq:heteo) with the squared noise, i.e. $z_{k,t} = q_k(\{\mathbf{z}_{t-\tau}\}) + \frac{1}{b_k(\{\mathbf{z}_{t-\tau}\})}\epsilon_{k,t}$.. We show the results of MCC trajectories of NPSSM in Figure A1. We can find that NPSSM still achieve high MCC around 0.9.

Table A5: Comparison between N-NPSSM and NPSSM for MCC scores and forecasting performance on synthetic datasets

| Method | Synthetic 1 | | | Synthetic 3 | | |
|---|---|---|---|---|---|---|
| | MCC | MAE | $R_{0.9}$-loss | MCC | MAE | $R_{0.9}$-loss |
| NPSSM | 0.984 | 0.073 | 0.288 | 0.933 | 0.061 | 0.384 |
| N-NPSSM | **0.987** | **0.054** | **0.220** | **0.998** | **0.057** | **0.363** |

| Method | Synthetic 2 | | | Synthetic 4 | | |
|---|---|---|---|---|---|---|
| | MCC | MAE | $R_{0.9}$-loss | MCC | MAE | $R_{0.9}$-loss |
| NPSSM | **0.996** | 0.080 | 2.178 | 0.946 | 0.095 | 0.196 |
| N-NPSSM | 0.995 | **0.069** | **1.866** | **0.992** | **0.081** | **0.169** |

Table A6: Ablation study for the effectiveness of the transition prior network

| Method | Synthetic 1 | Synthetic 2 | Synthetic 3 | Synthetic 4 |
|---|---|---|---|---|
| NPSSM w/t transition prior | 0.984 | 0.996 | 0.933 | 0.946 |
| NPSSM w/o transition prior | 0.641 | 0.700 | 0.729 | 0.634 |

### A3.3.4    MODEL COMPLEXITY

In Table A7, we report the total number of parameters of different methods in our synthetic experiments. Compared to baseline models, the proposed NPSSM and N-NPSSM requires more extra parameters. This is because these two methods have an extra transition prior models. Compared to NPSSM, N-NPSSM has more parameters since it needs to explicitly model the encoder for $\hat{\mathbf{c}}_{1:T}$ conditioned on $\{\hat{\mathbf{z}}_{t-\tau}\}_{\tau=0}^{L_c}$.

### A3.3.5    INTERPRETABILITY ANALYSIS

In Figure A2, we show the recovered causal relationships from NPSSM and KVAE in terms of MCC and causal-related factors in Synthetic3 dataset. Compared to KVAE, we can find that NPSSM has a better latent causal variables recovery, which is estimated up to permutation and component-wise invertible transformation.

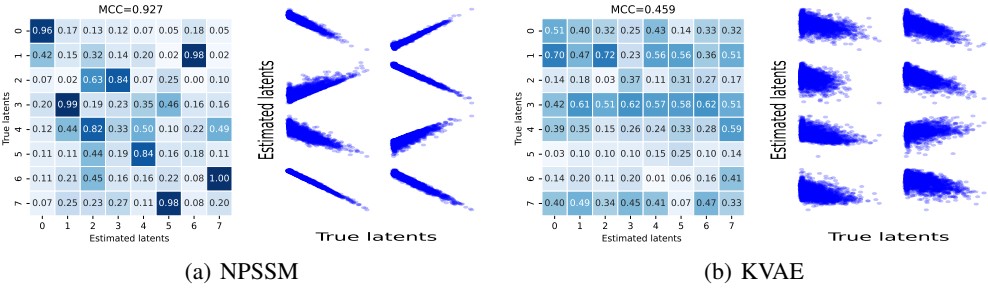

(a) NPSSM                    (b) KVAE

Figure A2: MCC for causally-related factors and scatter plots between estimated factors and true factors on two synthetic datasets for NPSSM.

Figure A3 present some showcases for different models in Economics dataset for qualitative evaluation. We can observe that N-NPSSM can predict well under various temporal data characteristics.

To visualize nonlinear relations, we use LassoNet (Lemhadri et al., 2021) as a post-processing tool to remove weak edges and generate the sparse causal relation graph from the results on the economics dataset. This method prunes input nodes by jointly feeding the first hidden layer and the residual layer through a hierarchical threshold-based optimizer. We first fit the LassoNet on the emission

Table A7: Model size (Total parameters) of different methods in synthetic experiments.

|  | LSTM | KVAE | VRNN | DeepAR | NPSSM | N-NPSSM |
|---|---|---|---|---|---|---|
| Model Size | 1k | 72.3k | 56.2k | 46.3k | 78.2k | 117k |

Figure A3: The observations of each model on economics dataset

model, which receives latent variables and outputs the observation variables at the same time step. As shown in Figure A4, we can find that industrial production and business confidence survey are simultaneously correlated, as both of them are affected by latent factor '1'. Additionally, foreign exchange reserves, CPI and money supply are simultaneously correlated, as all of them effected by latent factor '4'.

In Figure A5, we use LassoNet again to extract the sparse time-lagged causal relation in latent space. We can observe that most of the latent factors are affected by their time-lagged parents' nodes. Meanwhile, our model can also recover the cross relations between latent variables.

## A4 IMPLEMENTATION DETAILS

### A4.1 NETWORK ARCHITECTURE

We summarize our network architecture in Table A8.

### A4.2 TRAINING DETAILS

The models were implemented by `PyTorch` 1.9.0. The VAE network is trained using AdamW optimizer and early stops if ELBO loss does not decrease. The maximum epoch is 200 for synthetic datasets and 700 for real-world datasets. A mini-batch size of 64 is used. We used three random seeds in each experiment and reported the mean performance with standard deviation averaged across random seeds.

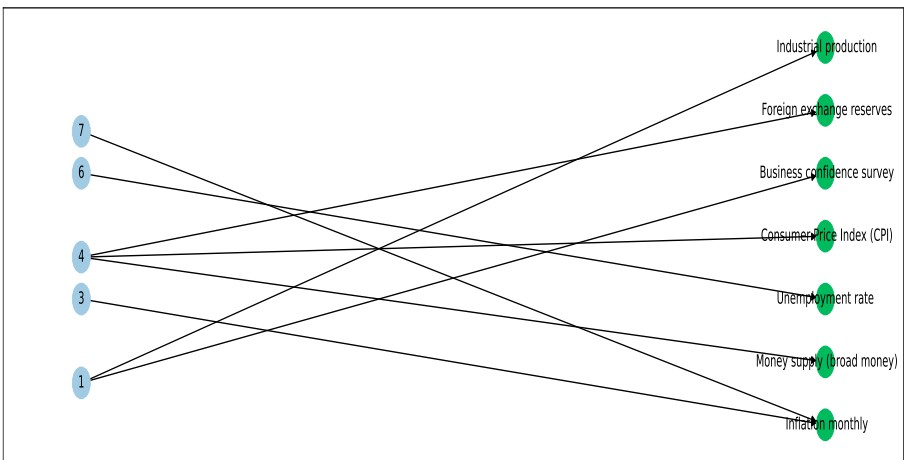

Figure A4: The causal relation between latent variables and observed variables. The blue circles with a number indicate latent factors, while the green circles indicate the observed variables. Note that latent factors '0', '2' and '5' has been removed by the pruning step when constructing this relation graph. It means these factors do not demonstrate strong causal strengths.

The hyperparameters of N-NPSSM include $[\beta, \gamma, \sigma]$, which represent the weights for transition prior for latent variable $\mathbf{z}$, change factor $\mathbf{z}$, and auxiliary predictor. Since the objective of transition prior does not consider the initial time-lagged variables, we follow the conventional VAE and use the standard normal distribution $\mathcal{N}(0, 1)$ as the prior distribution for these initial latent variables instead. Therefore, we augment the hyperparameters to $[\beta, \beta_{init}, \gamma, \gamma_{init}, \sigma]$. We performed a grid search to select these hyperparameters, which are $lr \in [1e-3, 5e-3, 2e-2]$, $\beta \in [8e-3, 1e-2, 2e-2]$, $\beta_{init} \in [5e-4, 2e-3]$, $\gamma \in [1e-4, 5e-3, 1e-2, 2e-2]$, $\gamma_{init} \in [3e-3, 5e-3, 2e-2]$, and $\sigma \in [0.1, 0.5, 1]$. To facilitate comparison, the training parameters of baselines, e.g., optimizer, batch size, as well as the encoder and decoder architecture are identical to N-NPSSM. Similarly, we performed a grid search to select learning rate, $lr \in [5e-4, 1e-3, 5e-3, 1e-2, 5e-2, 1e-1]$, and the hyperparameter of KL divergence term, $\alpha \in [1e-4, 5e-4, 1e-3, 5e-3, 1e-2, 5e-2, 1e-1]$. For all experiments, we use $\mathbf{z} \in \mathbb{R}^8$ and $\mathbf{c} \in \mathbb{R}^4$ and set the maximum time lag $L = 2$ by the rule of thumb. For the initialization of VAE, we follow the instruction of $\beta$-VAE (Higgins et al., 2016) and adopt the He initialization. For the rest of the modules/networks, we adopt uniform initialization.

**Training Stability** We have used several standard tricks to improve training stability: (1) we use AdamW optimizer as a regularizer to prevent training from being interrupted by overflow or underflow of variance terms of VAE; (2) For the experiments on synthetic datasets, we separate the learning procedure into two phases. We focus on the reconstruction task first and uncover the latent process, then we learn the latent predictor. This allows the model to first find the identifiable latent representations and then learn how to utilize them for the forecasting task. For the real-world datasets, we jointly learn these two components.

**Computation Hardware** We use Nvidia A100 GPU to run our experiments.

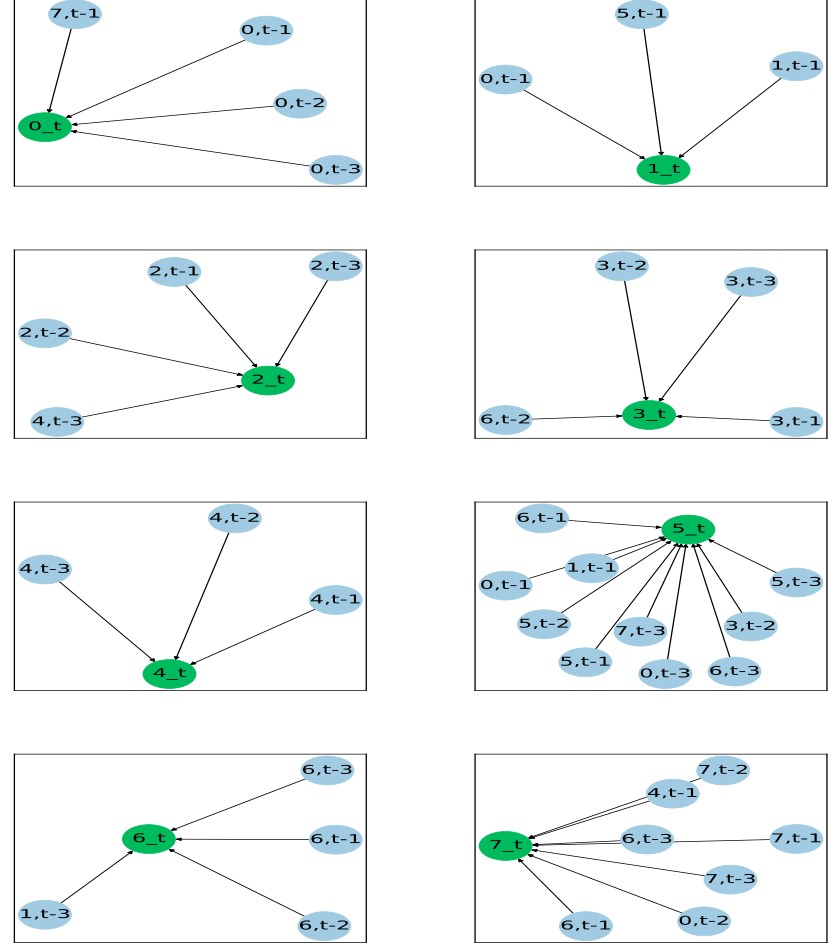

Figure A5: The time-lagged causal relations graph for latent variables. The blue circles indicate the time-lagged source latent factors, while the green circles indicate the target latent factors.

Table A8: Architecture details. BS: batch size, T: length of time series, i_dim: input dimension, z_dim: latent dimension, c_dim: time-varying change factor dimension, LeakyReLU: Leaky Rectified Linear Unit.

| Configuration | Description | Output |
|---|---|---|
| **1. MLP-Encoder** | Encoder Network | |
| Input: $\mathbf{x}_{1:T}$ | Observed time series | BS $\times$ T $\times$ x_dim |
| Dense | 128 neurons, LeakyReLU | BS $\times$ T $\times$ 128 |
| Dense | 128 neurons, LeakyReLU | BS $\times$ T $\times$ 128 |
| Dense | 128 neurons, LeakyReLU | BS $\times$ T $\times$ 128 |
| Dense | Temporal embeddings | BS $\times$ T $\times$ z_dim |
| **2. MLP-Decoder** | Decoder Network | |
| Input: $\hat{\mathbf{z}}_{1:T}$ | Sampled latent variables | BS $\times$ T $\times$ z_dim |
| Dense | 128 neurons, LeakyReLU | BS $\times$ T $\times$ 128 |
| Dense | 128 neurons, LeakyReLU | BS $\times$ T $\times$ 128 |
| Dense | i_dim neurons, reconstructed $\hat{\mathbf{x}}_{1:T}$ | BS $\times$ T $\times$ i_dim |
| **3. Inference Network for $\mathbf{z}_{1:T}$** | Inference Network | |
| Input | Temporal embeddings | BS $\times$ T $\times$ z_dim |
| Bottleneck | Compute mean and variance of posterior | $\mu_{1:T}^z, \sigma_{1:T}^z$ |
| Reparameterization | Sampling | $\hat{\mathbf{z}}_{1:T}$ |
| **4. Inference Network for $\mathbf{c}_{1:T}$** | Inference Network | |
| Input | Temporal embeddings | BS $\times$ T $\times$ c_dim |
| Bottleneck | Compute mean and variance of posterior | $\mu_{1:T}^c, \sigma_{1:T}^c$ |
| Reparameterization | Sampling | $\hat{\mathbf{c}}_{1:T}$ |
| **5. Transition Prior for $\mathbf{z}_{1:T}$** | Nonlinear Transition Prior Network | |
| Input | Sampled latent variable sequence $\hat{\mathbf{z}}_{1:T}$ and $\hat{\mathbf{c}}_{1:T}$ | BS $\times$ T $\times$ z_dim |
| InverseTransition | Compute estimated residuals $\hat{\epsilon}_{it}$ | BS $\times$ T $\times$ z_dim |
| JacobianCompute | Compute $\log\left(\left|\det\left(\mathbf{J}\right)\right|\right)$ | BS |
| **6. Transition Prior for $\mathbf{c}_{1:T}$** | Nonlinear Transition Prior Network | |
| Input | Sampled latent variable sequence $\hat{\mathbf{c}}_{1:T}$ | BS $\times$ T $\times$ c_dim |
| InverseTransition | Compute estimated residuals $\hat{\zeta}_{it}$ | BS $\times$ T $\times$ c_dim |
| JacobianCompute | Compute $\log\left(\left|\det\left(\mathbf{J}\right)\right|\right)$ | BS |
| **7. Auxiliary Predictor** | Prediction Network | |
| Input | Sampled latent variable sequence $\hat{\mathbf{z}}_{1:T}$ | BS $\times$ T $\times$ z_dim |
| LSTMInference | Use past $\{\hat{\mathbf{z}}_{t-\tau}\}$ to predict $\hat{\mathbf{z}}_t$ | BS $\times$ T $\times$ z_dim |

