# OpenReview forum: "Non-Parametric State-Space Models: Identifiability, Estimation and Forecasting"
_ICLR.cc/2023/Conference — Submitted to ICLR 2023_

### Official Review · Reviewer_qCMD · 2022-10-20

**Confidence:** 3
**Correctness:** 3
**Technical Novelty And Significance:** 2
**Empirical Novelty And Significance:** 2
**Recommendation:** 6

**Clarity, Quality, Novelty And Reproducibility:**

The clarity is fine, but can be improved. There are several typos and I list a few that I noted while reading.
* In Sec 4: “...variables are sparse an…” an -> and
* Paragraph before Sec 5.1, “We compare NPSSM with typical deep forecasting models
”. But in Table 1 and 2, N-NPSSM is used.

The quality of the work can be improved in both theoretical and empirical sides, especially on the experimental results as already mentioned in the weakness of the paper. On the theoretical side, I only have a minor comment. The proposed theorem is for NPSSM, but the main results in Table 1 and 2 are all for N-NPSSM (time varying version of NPSSM). Fixing this mismatch would strengthen the usefulness and impact of the theorem.

I think the novelty is also fine, it is a small step towards a more general and flexible SSM for forecasting tasks.

The reproducibility is fine for me, training and architecture details are provided in the appendix, but not the code.

**Strength And Weaknesses:**

The strength of the paper is the balanced contributions in theoretical, methodological and empirical aspects. It is a small step towards a more general and flexible SSM for forecasting tasks.

The weakness of the paper mostly is the experiment. Even though it seems strong empirically, the setting is not very convincing for me. Below is a list of comments on possible way to improve experiments.
* There are no results for NPSSM and include it may serve a good ablation study.
* The author mentioned many deep learning and deep state space models in the related work but 4 are chosen including a naive LSTM. Including more baselines for sure could help.
* Nowadays, 3 datasets are very limited from my perspective.
* Looking one step ahead is fine but would be better if multi-step ahead is also considered.
* If using one step ahead for evaluation, one simple baseline can be the last seen value.
* Why R_{0.9} loss instead of considering the average of all the quantiles?
* The hyperparameters and training detail are missing for the baselines.
* Regarding to the MCC in Table 1: Is it fair to compare latent space with other algorithms when the data generation largely follows the proposed architecture?


**Summary Of The Paper:**

This work proposed a new State Space Model (SSM) for forecasting. The proposed work relies on 1) non-parametric functional causal model (Pearl, 2009) for the transition process and 2) post-nonlinear model (Zhang & Hyvarinen, 2012) to capture nonlinear distortion effects in the emission model. The resulting model is named as NPSSM. They also extend it with time-varying change factors, which serves as input to the transition process, to model the non-stationary aspect (N-NPSSM).

The authors proved, under certain conditions, their proposed NPSSM is identifiable. The proof extends the theorem in Yao et al., 2022 to additive noise case with post non-linear function (their formulation of NPSSM). An estimation framework based on structural VAE is proposed and experiments on 4 synthetic datasets and 3 real-world datasets demonstrated superior performance over 4 deep-forecasting and deep state-space models.


**Summary Of The Review:**

The paper has balanced contributions in theoretical, methodological and empirical aspects. It is a small step towards a more general and flexible SSM for forecasting tasks. But the theoretical and methodological contributions are extensions of previous works, limiting the significance. More importantly, the experiments section need to be improved so that the empirical results are more convincing. Please refer to Strength and Weakness section for more detail.

=== After authors' response ===
The authors addressed my questions on the empirical side and I increased my score to 6.

---

> ### Author Response · Authors · 2022-11-17
> **response to reviewer qCMD (1)**
>
> We sincerely thank the reviewer for the dedicated time and the helpful comments. Below we address each comment individually.
>
> ***Q1***: This work is a small step towards a more general and flexible SSM for forecasting tasks. The theoretical and methodological contributions are extensions of previous works, limiting the significance
>
> ***A1***: Thanks for your comment; please let us share our view of the novelty of the work. First, compared to the existing work [Yao 2022], our identifiable analysis takes into account the observation noise in a flexible form, which is essential to an SSM. As you can see from the proof, the identifiability in this case is much harder. Second, our work builds a connection between recovering latent processes and time series forecasting–we explained why time series forecasting can naturally benefit from the SSM, by recovering the latent processes and exploiting their temporal predictability and demonstrated it with extensive empirical results. In contrast, [Yao 2022] only focuses on recovering latent processes.
>
>
> ***Q2***: Ablation study of NPSSM
>
> ***A2***: Thanks for your questions. Actually, we included this ablation study in appendix A2.3.2 in our initial version. In terms of identifiable performance, we can find that NPSSM achieve comparable performance to N-NPSSM on synthetic1 and synthetic2. Since the generation procedure for these two datasets do not include time-vary change factors. We can also find N-NPSSM performs better than NPSSM on synthetic3 and synthetic4. Since these two datasets include time-varying change factors and NPSSM does not have a mechanism to capture this pattern.
>
> ***Q3***: Including more baselines for deep SSM
>
> ***A3***: Thanks for your helpful suggestion. We have added more baselines about deep SSM (DKF[1], SRNN[2], RVAE[3],  DSAE[4]) based on the comprehensive review of deep SSM [5]. The results are demonstrated in the following table. We can find that the proposed model still outperforms these baselines. We think the updated empirical results are more convincing to validate the effectiveness of the proposed method.
>
> |      | Economics      |               | Bitcoin         |                | FRED           |                |
> |------|----------------|---------------|-----------------|----------------|----------------|----------------|
> |      | MAE            | R_{0.9}-loss  | MAE             | R_{0.9}-loss   | MAE            | R_{0.9}-loss   |
> | DKF  | 0.784 | 1.877 | 0.516  | 0.148 | 0.722 | 0.901 |
> | RVAE | 0.798 | 1.775 | 0.524  | 0.183 | 0.666 | 1.128 |
> | SRNN | 0.710 | 1.266 | 0.530  | 0.202 | 0.787 | 1.289 |
> | DSAE | 0.736 | 1.594 | 0.493 | 0.227 | 0.650 | 1.061 |
> | N-NPSSM | 0.603 | 1.190 |  0.403 | 0.143 | 0.484 | 0.580 |
>
>
> [1] Deep Kalman filters
> [2] Sequential neural models with stochastic layers
> [3] A recurrent variational autoencoder for speech enhancement
> [4] Disentangled sequential autoencoder
> [5] Dynamical Variational Autoencoders: A Comprehensive Review
>
> ***Q4***: Include more datasets
>
> ***A4***: Thanks for your suggestion. We add 3 more real-world datasets from various domain, including NN5 [1], OIKOLAB[2], Pedestrian[3] in our benchmark. The results are reported in the following table, we can find that N-NPSSM still outperforms the baselines.
>
> |         | NN5   |              | OIKOLAB |              | Pedestrain |              |
> |---------|-------|--------------|---------|--------------|------------|--------------|
> |         |   MAE | R_{0.9}-loss |     MAE | R_{0.9}-loss |        MAE | R_{0.9}-loss |
> | LSTM    | 0.890 |          N/A |   0.422 |          N/A |      0.739 |          N/A |
> | KVAE    | 0.573 |        8.858 |   0.424 |        1.190 |      0.362 |        6.475 |
> | VRNN    | 0.869 |        9.259 |   0.842 |        1.310 |      0.769 |        9.794 |
> | DeepAR  | 0.567 |        6.405 |   0.191 |        0.325 |      0.387 |        4.573 |
> | N-NPSSM | 0.560 |        5.969 |   0.152 |        0.323 |      0.293 |        4.191 |
>
> [1] https://zenodo.org/record/4656117
> [2] https://zenodo.org/record/5184708
> [3] https://zenodo.org/record/4656626
>
>
>
> ***Q5***: Add naive forecasting baselines
>
> ***A5***: Thanks for your suggestion. The results for naive forecasting are shown in the following table. We can find the proposed N-NPSSM can beat it on both synthetic and real-world datasets.
>
>
> |   MAE      | Synthetic1 | Synthetic2 | Synthetic3 | Synthetic4 | Economics | Bitcoin |  FRED  |
> |---------|:----------:|:----------:|:----------:|:----------:|:---------:|:-------:|:------:|
> |   Naive |    0.540    |    0.593   |   0.6615   |   0.6278   |   0.796   |  0.8765 | 0.791 |
> | N-NPSSM |      0.054 |      0.069 |      0.058 |      0.081 |     0.603 |   0.403 |  0.484 |

---

> > ### Author Response · Authors · 2022-11-17
> > **response to reviewer qCMD (2)**
> >
> > ***Q6***: Add the multi-step forecasting setting in experiments
> >
> > ***A6***: Thanks for your suggestion. The results for multi-step (3-step) forecasting are shown in the following table.  We can find that N-NPSSM outperforms the baselines consistently.
> >
> > | Multi-step | Economic        |                   | Bitcoin         |                   | Fred            |                   |   |
> > |------------|-----------------|-------------------|-----------------|-------------------|-----------------|-------------------|---|
> > |            | MAE | R_{0.9}-loss | MAE | R_{0.9}-loss | MAE | R_{0.9}-loss |   |
> > | LSTM       |           0.724 |             N/A |           0.634 |             N/A |           0.694 |             N/A |   |
> > | DeepAR     |           0.895 |             1.665 |           1.483 |             0.658 |           0.922 |             0.677 |   |
> > | KVAE       |           0.612 |             1.365 |           0.551 |             0.299 |           0.534 |             0.795 |   |
> > | VRNN       |           0.797 |             1.592 |           0.685 |             0.248 |           0.728 |             1.009 |   |
> > | N-NPSSM    |           0.588 |             1.177 |           0.430 |             0.166 |           0.595 |             0.602 |   |
> >
> >
> > ***Q7***: Add evaluation metric for the average of all the quantiles.
> >
> > ***A7***: Thanks for your suggestion. We follow [1] and use continuous ranked probability score (CRPS), which can be treated as an average of all the quantiles,  to measure the fit of the predictive distribution to the true one. The results are shown in the following table.
> >
> > | CRPS    | Economic | Bitcoin |  Fred |
> > |---------|---------:|--------:|------:|
> > | KVAE    |    1.231 |   0.301 | 0.892|
> > | VRNN    |    1.217 |   0.375 | 0.997 |
> > | DeepAR  |    1.338 |   0.768 | 1.026 |
> > | N-NPSSM |    1.093 |   0.199 | 0.875|
> >
> >
> >
> > [1] High-Dimensional Multivariate Forecasting with Low-Rank Gaussian Copula Processes
> >
> >
> > ***Q8***: Training details for the baselines, reproducibility
> >
> > ***A8***: Thanks for your comments. We have added the description in appendix A3.2
> >
> > ***Q9***: Is it fair to compare latent space with other algorithms when the data generation largely follows the proposed architecture?
> >
> > ***A9***: Thanks for your comments. First, we want to highlight that our method enjoys a more general form of SSM compared to other baselines, especially the non-additive form of transition model, in which all the baselines of deep SSM have not considered it. While the model is more general, the proposed model still relies on the assumptions in theorem1. We therefore conduct the experiments for violation of the critical assumption and explore the setting with (1) dependent noise and (2) additive Gaussian noise in Figure 3. We can find that the performance drops on these two settings. Inspired by your questions, to make our empirical analysis more thorough, we further consider the setting of instantaneous effect, whose latent generation process is defined as:
> > $z_\{k,t} = A z_t + q_k(\\{z_\{t-\tau}\\}) + \frac{1}{b_k(\\{z_\{t-\tau}\\})}\epsilon_\{k,t}.$
> >  We find the identifiability performance deterioration that the final MCC score is around 0.74.

---

### Official Review · Reviewer_Uowm · 2022-10-25

**Confidence:** 4
**Correctness:** 3
**Technical Novelty And Significance:** 3
**Empirical Novelty And Significance:** 3
**Recommendation:** 6

**Clarity, Quality, Novelty And Reproducibility:**

The clarity of the paper can be improved. In the present form, some of the critical technical pieces are unclear, especially regarding the role of the causal model in the SSM (see details above). As a result of this, it was difficult to truly assess the novelty of the presented method and the reproducibility also becomes limited.


**Strength And Weaknesses:**

Strengths:


The use of functional causal modeling in SSM is interesting and promising. The issues of dealing with non-additive noises and non-stationary transition functions are also important.



Weakness:

The writing of the papers is generally difficult to follow. It was not clear how some of the stated goals were achieved.

- The paper emphasizes "non-parametric" models, referring presumably the use of casual function modeling for the transition process. How is this model effectually reflected in the estimated SSM is quite unclear. As described in page 5-6, the latent prediction model p_pred of z essentially is modeled as a LSTM, and the noise is generated from the inverse latent transition function that is modeled as MLPs. It is not clear where is the latent causal model in the final ELBO loss in Equation (7). Is it in p(z_1:T) and p(c_1:T)? If yes, what are their relations to the "latent prediction model"? Why do we need both the latent causal model (equation 2) and latent prediction models at the same time? All of these questions made it difficult to assess the contribution of the presented causal transition functions.

- How is the 2nd term in the ELBO loss (7) supervised at training time? What's the significance of this term?

- The baselines used are limited. As revised in the related works, there is a large set of SSMs with complicated modeling of q(z) and p(z)'s, such as those below. These should be included in the comparisons in order to truly understand the contribution of the presented work over state of the arts.

Krishnan et al, Deep Kalman Filters, 2015

Klushyn et al, Latent matters: Learning deep state-space models, NeurIPS 2021


- Please elaborate on the definition and calculation of MCC in the main text. In the Appendix, it is stated that 'It first computes the absolute values of the Spearman’s rank
correlation coefficients between every ground-truth factor against every estimated latent factor". It is not clear how is this done in the temporal setting. What are ground-truth factors in the synthetic experiments and ho are the "estimated latent factor" determined? How are all the permutation done in a time series setting for all z_t's over time? How are these done for the baseline models?

- Given that the method introduced two rather complicated components to state-of-the-art SSMs, an ablation study is needed to understand the importance of each of the components (causal model, and non-additive noises)


- As the authors discuss identifiability, it seems that it mainly refers to the identifiability of the latent variables z_t's. Does the identifiability of z_t's directly translates to the identifiability of the underlying transition function? In traditional SSM, the identifiability of the system states and system parameters (that define the transition functions and others) are not necessarily the same?

**Summary Of The Paper:**

This paper presents a SSM adopting functional causal model for transition and post nonlinear model for emission, inferred in a VAE framework. The method was motivated to overcome several limitations of existing SSM formulations including the use of additive noises, and handling non-stationary time series. Experiments were conducted on synthetic and real-world datasets with a small number of selected baselines.

**Summary Of The Review:**

This paper discusses two interesting ideas for improving deep SSMs. However, the lack of clarity, the insufficient consideration of comparison models, and the lack of ablation study made it difficult to assess the actual impact of these two ideas on improving SSM-based time-series modeling and forecasting.

---

> ### Author Response · Authors · 2022-11-17
> **response to reviewer Uowm (1)**
>
> We sincerely thank the reviewer for the time devoted and the thorough comments, many of which are related to the novelty and key contributions of this work. It will help improve the readability of our work. Therefore, we attempt to address all the concerns in the following.
>
> ***Q1***: How is the proposed “non-parametric” model (latent causal model) effectually reflected in the estimated SSM is quite unclear.
>
> ***A1***: Thanks for asking this question, which helped improve our presentation. As we introduced in the paragraph for introducing VAE in Section 3, we use the transition prior $p(\hat{z}_\{1},\dots, \hat{z}_\{T}) = p(\hat{z}_\{1})\cdots p( \hat{z}_\{L})\prod^T_\{t=L+1}p(\hat{z}_t| \\{\hat{z}_\{t-\tau}\\}^L_\{\tau=1}, \hat{c}_t)$
> and $p(\hat{c}_1, \dots, \hat{c}_T) = p(\hat{c}_1)\prod^T_\{t=2} p(\hat{c}_t|\hat{c}_\{t-1})$  to encode latent causal relationships for uncovering the latent transition process. In the following paragraph, we explicitly introduce how we implement this transition prior model $p(\hat{z}_t| \\{\hat{z}_\{t-\tau}\\}^L_\{\tau=1}, \hat{c}_t)$ and $p(\hat{c}_t|\hat{c}_\{t-1})$.
> We have updated the presentation of  the prior in VAE in Section 3, to reflect this.
>
> ***Q2***: It is not clear where is the latent causal model in Equation (7). Is it in p(z_1:T) and p(c_1:T)? If yes, what are their relations to the "latent prediction model"? Why do we need both two models at the same time?
>
> ***A2***：Thanks for carefully reviewing the paper. Yes, uncovering the latent temporal process is implemented in the proposed transition prior model $p(z_1\cdots,z_T)$ and $p(c_1\cdots,c_T)$. The latent causal model is implemented by encoder, decoder and the transition prior model. It corresponds to the reconstruction term, KL divergence term for both $\hat{z}_\{1:T}$ and $\hat{c}_\{1:T}$.  While the latent causal model could uncover and estimate latent variable $z_t$, it could not explicitly model the forward prediction relationship which is required for the forecasting task. Therefore, we add an auxiliary latent prediction model, which can be treated as a regularization term in the objective function to enforce the temporal predictability of the learned latent processes. We further explain each specific design.
>
> In this work, we propose a flexible (non-parametric) form $z_t = f(\\{z_\{t-\tau}\\}^L_\{\tau=1}, c_t, \epsilon_t)$ (shown in Eq. (2)), which can handle various interactions among change factors $c_t$, cause variables $Parents(z_\{it})$ and noise variables $\epsilon_\{it}$. To achieve this, we argue that explicitly using this forward prediction function $f(\cdot)$ with fixed loss can not model the latent process without parametric forms. For example, the latent process in our synthetic data in Eq. 18-20, $z_\{k,t} = q_k(\\{z_\{t-\tau}\\}^L_\{\tau=1}) + \frac{1}{b_k(\\{z_\{t-\tau}\\}^L_\{\tau=1})}\epsilon_\{k,t}$ cannot be estimated by forward prediction function $f(\cdot)$ with squared loss because of the coupling effect from noise variable and cause variable. In contrast, as shown in Eq. 5, this transition probability can also be formulated as the product of noise probability and the determinant of the Jacobian matrix by applying the change of variables formula
> $$p\left(\hat{z}_\{it}|  \\{\hat{z}_\{t-\tau}\\}^L_\{\tau=1}, \hat{c}_\{t}\right) = p_\{\epsilon_\{it}}\left(r_i(\hat{z}_\{it}, \hat{c}_\{t}, \\{\hat{z}_\{t-\tau}\\}^L_\{\tau=1})\right)\left|\frac{\partial r_i}{\partial\hat{z}_\{it}}\right|$$
>
> We argue that this formulation enjoys several benefits. First, explicitly modeling the noise variable $\hat{\epsilon}_\{it}=r_i(\hat{z}_\{it}, \hat{c}_\{t},\\{\hat{z}_\{t-\tau}\\}^L_\{\tau=1})$  helps enforce the independent noise assumption from theorem 1. Since each noise is modeled with a separate MLP network. Second, as we discussed in the paragraph of introducing the transition prior model, since the Jacobian is a lower-triangular,  we can efficiently calculate its determinant as the product of each diagonal element. Given this, the transition probability $p(\hat{z}_t| \\{\hat{z}_\{t-\tau}\\}^L_\{\tau=1}, \hat{c}_t)$ can be efficiently evaluated using factorized noise distribution $\prod p(\epsilon_\{it})$.
>
> While the above method could estimate latent variable $z_t$ in the non-parametric form, **it does not explicitly model the temporal relationship which is required for the forecasting task**. Thus, we further propose the auxiliary latent predictor $p_\{pred}(\hat{z}_\{t}|\hat{\epsilon}_\{t}, \\{\hat{z}_\{t-\tau}\\}^\{t-1}_\{\tau=1},  \hat{\epsilon}_\{t})$  by using the recovered latent processes $\hat{z}_1, \dots, \hat{z}_T$. With penalization parameter $\sigma$, one can view this term as a regularization term in the objective to enforce the temporal predictability of the learned latent processes.
>
> According to your question, we have improved our writing by adding more explanation of our estimation framework.

---

> > ### Author Response · Authors · 2022-11-17
> > **response to reviewer Uowm (2)**
> >
> > ***Q3***: How is the 2nd term in the ELBO loss (7) supervised at training time? What's the significance of this term?
> >
> > ***A3***: As we introduced below Eq. (7),  given the recovered latent variables $\\{\hat{z}_t\\}^T_\{t=1}$, $p_\{pred}(\hat{z}_\{t}|\hat{\epsilon}_\{t}, \\{\hat{z}_\{t-\tau}\\}^L_\{\tau=1})$ is optimized via MSE loss. As mentioned in the response to Q2, with penalization hyperparameter $\sigma$ , one can view this term as a regularization term in the objective in Eq. (7) to enforce the temporal predictability of the learned latent processes (it is not supervised), for the purpose of time series forecasting.
> >
> >
> > ***Q4***: Add more baselines about SSM discussed in related work sections
> >
> > ***A4***: Thanks for your helpful suggestion. We have added more baselines (DKF[1], SRNN[2], RVAE[3],  DSAE[4]) based on your suggestion and the survey of deep SSM [5].  For the work you mentioned [6], it is not a new Deep SSM but an optimization framework for general deep SSM. We do not find any public source code and contact the authors but they do not share the code. We have tried our best to reproduce it, but as the reviewer mentioned [7] “ the overall framework is fairly complex and hence difficult to implement from scratch”. The results are demonstrated in the following table. We can find that the proposed model still outperforms these baselines. We think the updated empirical results could further verify the effectiveness of the proposed model.
> >
> > |      | Economics      |               | Bitcoin         |                | FRED           |                |
> > |------|----------------|---------------|-----------------|----------------|----------------|----------------|
> > |      | MAE            | R_{0.9}-loss  | MAE             | R_{0.9}-loss   | MAE            | R_{0.9}-loss   |
> > | DKF  | 0.784 | 1.877 | 0.516  | 0.148 | 0.722 | 0.901 |
> > | RVAE | 0.798 | 1.775 | 0.524  | 0.183 | 0.666 | 1.128 |
> > | SRNN | 0.710 | 1.266 | 0.530  | 0.202 | 0.787 | 1.289 |
> > | DSAE | 0.736 | 1.594 | 0.493 | 0.227 | 0.650 | 1.061 |
> > | N-NPSSM | 0.603 | 1.190 |  0.403 | 0.143 | 0.484 | 0.580 |
> >
> > [1] Deep Kalman filters
> >
> > [2] Sequential neural models with stochastic layers
> >
> > [3] A recurrent variational autoencoder for speech enhancement
> >
> > [4] Disentangled sequential autoencoder
> >
> > [5] Dynamical Variational Autoencoders: A Comprehensive Review
> >
> > [6] Latent Matters: Learning Deep State-Space Models
> >
> > [7] https://openreview.net/forum?id=-WEryOMRpZU
> >
> > ***Q5***: Please elaborate on the definition and calculation of MCC. It is not clear how is this done in the temporal setting. What are ground-truth factors in the synthetic exp and how are the "estimated latent factor" determined? How are all the permutation done in a time series setting for all z_t's over time? How are these done for the baseline models?
> >
> > ***A5***: Thanks for your question. We focus on how well the underlying latent process $z_{t}$ are recovered in this evaluation procedure, and we didn't take into account the temporal structure of each $z_{it}$. We will explain the reason in Q7/A7.
> >
> > Specifically, we present the synthetic data generation procedures in Appendix A2.1.1 with different settings. Given this, we generate the data $\\{x_t, z_t\\}^T_\{t=1},$, where $\\{x_t\\}^T_\{t=1}$ are observed samples and $\\{z_t\\}^T_\{t=1}$ are the ground truth latent factors at $t$. Our model and the baselines w.r.t. SSM use the observation data $\\{x_t\\}^T_\{t=1}$ to obtain the estimated latent factors  $\\{\hat{z}_t\\}^T_\{t=1}$. In our model, this is calculated by the posterior (encoder) $\\{q(\hat{z}_t|x_t)\\}^T_\{t=1}$. To measure the performance of identifiability, we measure the discrepancy Mean Correlation Coefficient (MCC) between $\hat{z}_t$ and $z_t$ at each timestamp $t$ for our models and baselines.
> >
> > As shown in definition 1 and theorem1, we consider the identifiability up to relative minimum indeterminacies. That is, $z_t$  is a component-wise transformation of a permuted version of $\hat{z}_t$. Accordingly, we adopt the following procedure to calculate the MCC. We first apply a nonlinear regression to the recovered factors, aiming to get rid of the component-wise transformation indeterminacy, for each possible pair of the estimated factor and the true one. Then, we calculate all pairs of correlation coefficients (the absolute values of the Spearman’s rank correlation coefficients) between ground-truth latent factors and the estimated latent factors (after the component-wise transformation). We further solve a linear sum assignment problem to assign each latent component to the ground-truth component that best correlates with it, thus finding the correspondence between the estimated factors and the true ones in the latent space. A high MCC means one successfully recovered the true latent factors, up to invertible, component-wise transformation and permutation.
> >
> > Thanks for pointing this out. We have improved the writing for clarifying MCC calculation in both the main context and appendix.

---

> > > ### Author Response · Authors · 2022-11-17
> > > **response to reviewer Uowm (3)**
> > >
> > > ***Q6***: Given that the method introduced two rather complicated components to state-of-the-art SSMs, an ablation study is needed to understand the importance of each of the components (causal model, and non-additive noises)
> > >
> > > ***A6***: Thanks for your suggestion. We have conducted this ablation study with/without transition prior module. The results are shown in the following table. We can find that this module is critical for recovering the latent variables, as we can see MCC scores increase significantly when using transition prior module.
> > >
> > > |             MCC            | Synthetic1 | Synthetic2 | Synthetic3 | Synthetic4 |
> > > |:--------------------------:|------------|------------|------------|------------|
> > > | NPSSM w/t transition prior | 0.984      | 0.996      | 0.933      | 0.94６     |
> > > | NPSSM w/o transition prior | 0.641      | 0.700      | 0.729      | 0.634      |
> > >
> > >
> > >
> > > ***Q7***: As the authors discuss identifiability, it seems that it mainly refers to the identifiability of the latent variables z_t's. Does the identifiability of z_t's directly translates to the identifiability of the underlying transition function? In traditional SSM, the identifiability of the system states and system parameters (that define the transition functions and others) are not necessarily the same?
> > >
> > > ***A7***: Thanks for this insightful comments. We have discussed this problem in the paragraph above definition 1. We agree that the identifiability of SSM including both system states and transition functions. Our Theorem 1 focus on the identifiability of latent variables, but as we explained before definition 1, once the latent variables $z_1, \dots, z_T$ are identifiable up to componentwise transformations and permutation, latent transition (causal relationships) are also identifiable qualitatively because conditional independence relations fully characterize time delayed causal relations in a time-delayed causally sufficient system. In fact, given the definition in Eq. (3), there is no latent confounder for $z_\{it}$ and parents nodes of $z_\{it}$ or instantaneous causal relations between $z_\{it}$ and $z_\{jt}$, for any $i,j$; as a consequence, in this case, causal structure information is fully recoverable from conditional independence relations (e.g., see the PC algorithm in Spirtes et al. (1993))
> > >
> > > At the same time, the quantitative causal models for $z_\{it}$ will be naturally sensitive to their componentwise transformations; such an indeterminacy may be eliminated when we have additional information, say, the distribution of each $z_\{it}$.
> > >
> > > We have updated our presentation to reflect it.

---

> > > > ### Comment · Reviewer_Uowm · 2022-11-30
> > > > **Thanks for the response**
> > > >
> > > > I want to thank the authors for very detailed and substantial response. It helped with better understanding of the work, and the added SSM baselines and ablation study also added more empirical evidence of the contribution of the work.
> > > >
> > > > It'd be great if the authors can further clarify two questions.
> > > >
> > > > 1. The contribution of the "auxiliary latent prediction model" is still not very clear to me. Indeed it works as a temporal regularization, but since it is to be learned anyway (i.e., it is not a known constraint) and it is very weakly supervised (only by the estimated zt which it is supposed to constrain), it is not clear how well this model can be learned and what its effect could be. It'd be good to see an ablation on this component as well (which was mentioned in my previous review).
> > > >
> > > > 2. I'm still not very clear about where the functional causal model ended up in the final model since, as explained by the authors, function f_i (or f_c) in equation (4) is not being used in the "transition prior model". Instead the nose function is being modeled using a set of MLP networks. Is the MLP fully data-driven, or somehow it encodes the causal relationship among z's? e.g., Are the parents of z_it somehow predefined? It in general is not crystal clear to me where the causal model is within the MLP models of the noise functions r_i's. It'd be helpful if the authors could clarify.
> > > >
> > > > 3. The authors also mentioned that " we argue that forward prediction with fixed loss cannot model latent processes without parametric form". What is the rationale for not learning a parametric form of f_i (with some forms of neural network) that includes the causal relations? That way one would also no longer needs the auxiliary latent predictor anymore. This is related to my question 2 -- fundamentally, the motivation to avoid modeling f_i, but modeling the noise, using neural networks.
> > > >
> > > > Thanks very much!

---

> > > > > ### Author Response · Authors · 2022-12-01
> > > > > **further response to reviewer Uowm (1)**
> > > > >
> > > > > Thanks for taking the time and effort for the review and response. Below we address each comment individually.
> > > > >
> > > > > ***Q8***: The contribution of the "auxiliary latent prediction model" is still not very clear. Ablation study for the predictor component.
> > > > >
> > > > > ***A8***: Thanks for your insightful suggestion. We show the ablation study in the following table on synthetic1 dataset.  The contribution of auxiliary latent prediction model is not to recover latent variables $z$ but to learn the forward prediction relationship.
> > > > >
> > > > > To address your concern, we present two solutions to optimize the overall objective function $\mathcal{L}$ in Eq. (7) ( We present more details of our training procedure in the paragraph of training stability in Appendix A4.2).
> > > > >
> > > > > * (1) two-phase learning, we first focus on recovering latent process $\\{z_t\\}^T_\{t=1}$ by only optimizing the reconstruction loss and the KL divergence term. Then, we freeze all the parameters (encoder, decoder, transition prior model) and only learn the auxiliary latent prediction model via the recovered latent variables $\\{\hat{z}_t\\}^T_\{t=1}$. Thus, minimizing the prediction error on the recovered latent variables $\\{\hat{z}_t\\}^T_\{t=1}$ is aligned with the goal of minimizing the prediction error on observations $\\{x\\}^T_\{t=1}$ since the encoder and decoder are fixed.  Besides, the recovered latent variables and MCC results will not change in this phase. From the following table we can see MCC is very high which indicates we successfully recover the latent process. Therefore, we do not agree estimated $z_t$ is a weakly supervised information.
> > > > >
> > > > > *  (2)  We joint learn the latent process (reconstruction error + KL divergence term) and the auxiliary latent prediction model. It is more challenging than the two-phase learning solution, but enjoys some extra benefits, like less effort for hyper-parameter tuning. Empirical results in the following table show that both the identifiability performance and forecasting performance are slightly worse than the results of two-phase learning, but the MCC is still very high which indicates we successfully recover the latent process and MAE is lower than other baselines which indicates a better forecasting performance.
> > > > >
> > > > > | NPSSM               | MCC   | MAE   |
> > > > > |---------------------|-------|-------|
> > > > > | two-phase learning  | 0.984 | 0.073 |
> > > > > | joint learning | 0.964 | 0.078 |
> > > > >
> > > > > ***Q9***: It is not clear where the causal model is within the noise function $r_i$.
> > > > >
> > > > > ***A9***: Thanks for your insightful questions. The parents of $z_{i,t}$ is not predefined, we use the maximum time lag $L$ as a pre-defined hyperparameter to reduce search space. As we explained in Q7, since our framework is able to identify the latent variables $z_1, \dots, z_T$, and we assume the time-delayed causally sufficient system (it indicates that there is no latent confounder for $z_{it}$ and parents nodes of $z_{it}$ or instantaneous causal relations between $z_{it}$ and $z_{jt}$, for any $i,j$), then we can apply conditional independent test (e.g., see the PC algorithm in Spirtes et al. (1993)) to fully recover the latent transition (causal relationships, parents of target node $z_{i,t}$).
> > > > >
> > > > > Noise function (MLP) $\hat{\epsilon}_\{i,t}=r_i(\hat{z}_\{it}, \hat{c}_\{t}, \\{\hat{z}_\{t-\tau}\\}^L_\{\tau=1})$ is fully data-driven, it implicitly encodes the information from our proposed non-parametric transition function  $z_t = f(\\{z_\{t-\tau}\\}^L_\{\tau=1}, c_t, \epsilon_t)$. Let me explain it step by step how it works.
> > > > >
> > > > > First, ***the definition itself has already encodes the information that the noise $\hat{\epsilon}_\{it}$ is mutually independent (i.e., spatially and temporally independent)***, it is sufficient to estimate $\hat{\epsilon}_\{it}$  with $\hat{z}_\{i,t}, \hat{c}_\{t}, \\{\hat{z}_\{t-\tau}\\}^L_\{\tau=1}$ (i.e. there is no need to take other variables, like $\hat{z}_\{t+1}$ as the input of noise function).
> > > > >
> > > > > Second, the (transition) prior is $p(\hat{z}_1,\dots, \hat{z}_T) = p(\hat{z}_1) \cdots p( \hat{z}_\{L})\prod^T_\{t=L+1} p(\hat{z}_t | \\{\hat{z}_\{t-\tau}\\}^L_\{\tau=1}, \hat{c}_t)$. For each distribution $p(\hat{z}_t| \\{\hat{z}_\{t-\tau}\\}^L_\{\tau=1}, \hat{c}_t)$, we design the following transformation: $\bf{A} \rightarrow \bf{B}$:
> > > > >
> > > > > $\underbrace{
> > > > > \begin{bmatrix}
> > > > > \hat{z}_\{t-L},
> > > > > \cdots,
> > > > > \hat{z}_\{t-1},
> > > > > \hat{z}_\{t}
> > > > > \end{bmatrix}^\{\top}
> > > > > }_\{\mathbf{A}},  \text{mapped to }
> > > > > \underbrace{
> > > > > \begin{bmatrix}
> > > > > \hat{z}_\{t-L},
> > > > > \cdots,
> > > > > \hat{z}_\{t-1},
> > > > > \hat{\epsilon}_\{t}
> > > > > \end{bmatrix}^\{\top}
> > > > > }_\{\mathbf{B}}.
> > > > > $
> > > > >
> > > > > We can find that its Jacobian is lower-triangular,
> > > > > $
> > > > > \mathbf{J}_\{\mathbf{A} \rightarrow \mathbf{B}} =
> > > > > \begin{pmatrix}\mathbb{I}_\{nL} & 0 \\\ \star &   \text{diag}\left(\frac{\partial r_i}{\partial \hat{z}_\{it}}\right)\end{pmatrix}
> > > > > $,
> > > > > where $n$ is the dimension.

---

> > > > > > ### Author Response · Authors · 2022-12-01
> > > > > > **further response to reviewer Uowm (2)**
> > > > > >
> > > > > > This is because ***we encode the information from non-parametric transition function $f$ that only historical variables can be the cause of the future variable (time-delayed causal relationship) and noise variables $\hat{\epsilon}_\{t}$ are independent of parents nodes $\hat{z}_\{t-L}, \cdots, \hat{z}_\{t-1}$***. By apply the change of variables, we can obtain the joint distribution of the latent variables as:
> > > > > >
> > > > > > $
> > > > > > \log p(\underbrace{\hat{z}_\{t-L}, \cdots, \hat{z}_\{t-1}, \hat{z}_\{t}}_\{\bf{A}} | \hat{c}_\{t} ) = \log p ( \underbrace{\hat{z}_\{t-L}, \cdots, \hat{z}_\{t-1}, \hat{\epsilon}_\{t}}_\{B} |  \hat{c}_\{t} ) +  \log \left(\lvert \det \left(\mathbf{J}_\{\mathbf{A} \rightarrow \mathbf{B}}\right) \rvert \right)
> > > > > > $
> > > > > >
> > > > > > ***We again encode the information that noise variables $\hat{\epsilon}_\{t}$ are independent of parents nodes $\hat{z}_\{t-L}, \cdots, \hat{z}_\{t-1}$  and each noise variable $\hat{\epsilon}_\{i,t}$  is mutually independent.*** Then, we have:
> > > > > >
> > > > > > $
> > > > > > \log p(\underbrace{\hat{z}_\{t-L}, \cdots, \hat{z}_\{t-1}, \hat{z}_\{t}}_\{\bf{A}} |  \hat{c}_\{t}) = \underbrace{\log p \left(\hat{z}_\{t-L}, \cdots, \hat{z}_\{t-1} \right) + \sum_\{i=1}^n \log p(\hat{\epsilon}_\{i,t} |  \hat{c}_\{t} )}_\text{Independent Noise Condition} + \log \left(\lvert \det \left(\mathbf{J}_\{\mathbf{A} \rightarrow \mathbf{B}}\right) \rvert \right)
> > > > > > $
> > > > > >
> > > > > > Then, we move the first term of RHS to the LHS and get the transition prior
> > > > > >
> > > > > > $
> > > > > > \log p\left(\hat{z}_t \vert \\{\hat{z}_\{t-\tau}\\}_\{\tau=1}^L,  \hat{c}_\{t}\right) = \sum_\{i=1}^n \log p(\hat{\epsilon}_\{i,t} | \hat{c}_\{i,t})+ \sum_\{i=1}^n \log \Big| \frac{\partial r_i}{\partial \hat{z}_\{it}}\Big|
> > > > > > $,
> > > > > >
> > > > > > where we can ***efficiently compute the determinant of Jacobian as the product of each diagonal element, since the Jacobian is a lower-triangular***. Finally, for each target variable $\hat{z}_\{i,t}$ we have
> > > > > >
> > > > > > $\log p\left(\hat{z}_\{it}|  \\{\hat{z}_\{t-\tau}\\}^L_\{\tau=1}, \hat{c}_\{t}\right) =  \log p_\{\epsilon_\{i,t}}\left(r_i(\hat{z}_\{it}, \hat{c}_\{t}, \\{\hat{z}_\{t-\tau}\\}^L_\{\tau=1})\right) + \log \left|\frac{\partial r_i}{\partial\hat{z}_\{it}}\right|$,
> > > > > >
> > > > > > It means that we can exactly infer the prior distribution $p\left(\hat{z}_\{it}|  \\{\hat{z}_\{t-\tau}\\}^L_\{\tau=1}, \hat{c}_\{t}\right)$ with the noise distribution $p_\{\epsilon_\{it}}\left(r_i(\hat{z}_\{it}, \hat{c}_\{t}, \\{\hat{z}_\{t-\tau}\\}^L_\{\tau=1})\right) $.
> > > > > >
> > > > > > According to your question, we will put the detailed derivation in the appendix in the next version.
> > > > > >
> > > > > >
> > > > > > ***Q10***: the motivation to avoid modeling f_i, but modeling the noise, using neural networks.
> > > > > >
> > > > > > ***A10***: The reasons for not learning a parametric form of $f$ but estimating noise distribution $p(\hat{\epsilon}_\{it})$ to exactly infer the transition prior distribution $p\left(\hat{z}_\{it}| \\{\hat{z}_\{t-\tau}\\}^L_\{\tau=1}, \hat{c}_\{t}\right)$ are:
> > > > > >
> > > > > > * First, in this work, we consider a flexible non-parametric transition function $f$. It indicates that we ***don't make any assumptions for the specific form of noise density and how the noise variable interacts with parents node variable, change factor variable***. If you make additive Gaussian noise assumption, it will be contrary to our original intention. Therefore, even we consider a parametric form of $f$, we still need the effort to estimate the noise density without a specific form.
> > > > > >
> > > > > > * Second, we assume the noise $\hat{\epsilon}_\{i,t}$ is mutually-independent, and it is independent of the parents node of $\hat{z}_\{i,t}$. We argue that ***our method is much easier to enforce these independence assumptions into transition prior (as we explained in the response of previous question)***  than the method of directly modeling a parametric form of $f$.
> > > > > >
> > > > > > * Third, because of the (conditional) independence condition, the Jacobian is a lower-triangular, we can ***efficiently calculate the Jacobian’s determinant as the product of each diagonal element***.
> > > > > >
> > > > > > * Fourth, we argue that most of existing end-to-end deep forecasting model implicitly includes two tasks: estimating the latent variables and building the predictor. ***Our innovative work explicitly decouple these two tasks by first learning the underlying latent temporal processes and then building the prediction model on the uncovered latent variables.  We explain the benefits of this new learning paradigm (we present it in the last paragraph of page 2 and figure 1 ) and verify it on both synthetic and real-world data***.

---

> > > > > > > ### Comment · Reviewer_Uowm · 2022-12-01
> > > > > > > **One last thing**
> > > > > > >
> > > > > > > Thanks for the response. Very helpful.
> > > > > > >
> > > > > > > Re: the ablation and the effect of the latent prediction model, I was mainly unclear about why the latent prediction model is absolutely necessary. Is forecasting done by this model alone? Does this model receive any training signals in terms of forecasting at training time? Or, at training time, the encoder-decoder does the reconstruction and recover latent variables, and the latent prediction model is only supervised by the recovered latent variables. Then at test time, the latent prediction model alone is used for forecasting?
> > > > > > >
> > > > > > > Thanks!

---

> > > > > > > > ### Author Response · Authors · 2022-12-02
> > > > > > > > **Thanks for your timely response!**
> > > > > > > >
> > > > > > > >
> > > > > > > > Dear Reviewer,
> > > > > > > >
> > > > > > > > Thanks for your timely feedback. Here is our response to your follow-up questions.
> > > > > > > >
> > > > > > > > Q: Is forecasting done by this model alone?
> > > > > > > >
> > > > > > > > A: Yes, only auxiliary latent prediction focus on the forecasting task.
> > > > > > > >
> > > > > > > > Q: Does this model receive any training signals in terms of forecasting at training time? Or, at training time, the encoder-decoder does the reconstruction and recovers latent variables, and the latent prediction model is only supervised by the recovered latent variables.
> > > > > > > >
> > > > > > > > A: The latter one is absolutely right!
> > > > > > > >
> > > > > > > > Q: Then at test time, the latent prediction model alone is used for forecasting?
> > > > > > > >
> > > > > > > > A: No. We still need the encoder to project historical observations into historical latent factors and the decoder to transform the predicted latent factor into prediction value in the observation space (we present the 3-step prediction procedure in the last paragraph of introducing auxiliary latent prediction model in Section 3).
> > > > > > > >
> > > > > > > > For example, if we want to predict $x_3$ given $x_1, x_2$, we first got $z_1 <- \text{Encoder} (x_1), z_2 <- \text{Encoder}(x_2)$, then we make prediction in latent space $z^\{Predict}_3 <- \text{LatentPredictor}(z_1, z_2)$, finally, we transform it back to observation space $x^\{Predict}_3 <- \text{Decoder}(z^\{Predict}_3)$

---

> > > > > > > > > ### Comment · Reviewer_Uowm · 2022-12-02
> > > > > > > > > **Thank you**
> > > > > > > > >
> > > > > > > > > Thanks for the  clarification. The discussion and revision have addressed my main questions about this work. I'll be raising my score in support. Thanks for a great discussion.

---

> > > > > > > > > > ### Author Response · Authors · 2022-12-02
> > > > > > > > > > **Thanks for your time and effort!**
> > > > > > > > > >
> > > > > > > > > > Dear reviewer Uowm,
> > > > > > > > > >
> > > > > > > > > > Thank you again for your valuable comments. According to your comments and discussions, we will update our manuscript in the next version. We really appreciate your efforts in reviewing our work and updating your score.

---

> > > > > > > > > > > ### Author Response · Authors · 2022-12-05
> > > > > > > > > > > **Thanks again for your time and efforts**
> > > > > > > > > > >
> > > > > > > > > > > Dear reviewer Uowm,
> > > > > > > > > > >
> > > > > > > > > > > Once again, we appreciate your remarkable engagement in the review process, your pertinent questions, and the fruitful discussion. We benefited much from your input and are grateful for the statement in your review that "this paper discusses two interesting ideas for improving deep SSMs."  ***If the discussion and updated manuscript address your concerns***, we are just humbly wondering whether you would like to update your recommendation to "accept" from "marginally above the acceptance threshold". Hope you will not find this message inappropriate, and thank you for your support.
> > > > > > > > > > >
> > > > > > > > > > > Best regards,
> > > > > > > > > > >
> > > > > > > > > > > Authors of #2303

---

> ### Author Response · Authors · 2022-11-28
> **response to Uowm**
>
> Dear reviewer Uowm
>
> Thanks for your great efforts in reviewing our paper. We have provided answers to your questions and revised the paper following your suggestions. Please kindly let us know if our response has addressed your concerns.
>
> Thanks again for your valuable review. Looking forward to your reply

---

### Official Review · Reviewer_AS1z · 2022-10-27

**Confidence:** 2
**Correctness:** 3
**Technical Novelty And Significance:** 3
**Empirical Novelty And Significance:** 3
**Recommendation:** 6

**Clarity, Quality, Novelty And Reproducibility:**

The introduction and the motivation of the work are clear. The novelty of the work is unclear since it is difficult to ascertain the utility of the change factors.

**Strength And Weaknesses:**

1. It is not clear how the ‘time-varying change factors’ c_t enter into the transition equations. Specifically, they are not present in the latent predictor model, nor in the loss function (apart from the KL). It is unclear if they are being used at all, and if so, how.
2. The metrics of comparison for the true and estimated latent variables can be much improved. The authors can learn an optimal permutation to compare the latent variables.
3. The invertibility of the transmission plays an important role in the proposed model: the model seems to be under an assumption that the transition is invertible. Please discuss the situation where the transition and change transition functions are not invertible. This would seemingly lead to errors in the inference.

Minor
1. In Figure 4, the readers are not able to know the order of the latents in scatter plots, in particular, which one is the latent 2 in the MCC matrix?
2. Some notations in equations need clarifying, e.g., ‘i’ and ‘j’ in Eq 2 need to be clearer, the difference between z_t and \hat z_t in Eq 7 should be clarified, the notations in A.3.2, and so on.
3. Grammar: page2, paragraph 2, line 3,  ‘, In particular’.


**Summary Of The Paper:**

The authors proposed a general formulation of SSMs, called the Non-Parametric State-Space Model (NPSSM). This framework is able to depict the non-stationarity of the latent process over time. Based on the identifiability of time-lagged latent variables, they proposed a new structural VAE for model estimation and forecasting tasks. They optimize the reconstruction of the observation and the prediction of the latent from the latent at the previous time.

**Summary Of The Review:**

The work is interesting, but the clarity of the methods needs work.

---

> ### Author Response · Authors · 2022-11-17
> **response to reviewer AS1z (1)**
>
> We greatly appreciate the reviewer for the dedicated time and the helpful comments. Below we give a point-by-point response to the comments.
>
> ***Q1***: The introduction and the motivation of the work are clear. The novelty of the work is unclear
>
> ***A1***: Thanks for your comments. The novelty of the work consists of three parts. First, while we have witnessed the success of recent work on incorporating deep neural networks with SSM, most (if not all) of them follow the additive-noise form, which may not hold in practice. This limits the applicability of SSM in real applications. To the best of our knowledge, we are the first to propose such a general form of the emission and transition models in the non-parametric way (together with a higher-level model to capture time-varying change, if needed), while still enjoying identifiability guarantees, as discussed below.
>
> Second, identifiability of SSM is an important but challenging problem. However, most of the existing identifiability analyses focus on linear SSM while we have witnessed a lot of recent work investigating deep SSM . In this work, we first present the identifiability analysis for deep SSM, even the transition and emission model are more general than existing works.
>
> Third, in the view of time series forecasting, instead of building an end-to-end deep forecasting, we first propose a new learning paradigm by first learning the underlying latent temporal processes and then building the prediction model on the uncovered latent variables. We explain the benefits of this new learning paradigm and verify it on both synthetic and real-world data. We believe that our work will be beneficial to the Deep SSM community, identifiability analysis of SSM community and time series forecasting community.
>
> ***Q2***：It is not clear how the ‘time-varying change factors’ c_t enter into the transition equations. Specifically, they are not present in the auxiliary latent predictor model, nor in the loss function (apart from the KL). It is unclear if they are being used at all, and if so, how.
>
> ***A2***: Thanks for raising this point. It is related to the novelty and key contributions of our work. As you said, change factor c_t are not present in the reconstruction error and latent predictor, but the KL divergence. Let us explain it step by step (and include a summary in the updated manuscript).
>
>
> First, from the formulation of N-NPSSM in Eq. (4), we can observe that $x_t$ only depends on $z_t$, and thus the loss function (reconstruction error) is only related to $z_t$ not $c_t$.
>
> Second, as introduced in the paragraph of VAE in section 3, we use the transition prior $p(\hat{z}_1,\dots, \hat{z}_T)= p(\hat{z}_1)\cdots p( \hat{z}_L)\prod^T_\{\tau=1}p(\hat{z}_t | \\{\hat{z}_\{t-\tau}\\}^L_\{\tau=1}, \hat{c}_t)$ and $p(\hat{c}_1, \dots, \hat{c}_T) = p(\hat{c}_1)\prod^T_\{t=2} p(\hat{c}_t|\hat{c}_\{t-1})$  to encode latent causal relationships for uncovering the latent transition process. Therefore, $c_t$ are present in the prior. To support more general non-parametric transition process in estimation procedure, we formulate the transition prior as the product of noise probability and the determinant of the Jacobian matrix by applying the change of variables formula, $$p\left(\hat{z}_\{it}| \\{\hat{z}_\{t-\tau}\\}^L_\{\tau=1}, \hat{c}_\{t}\right) = p_\{\epsilon_\{it}}\left(r_i(\hat{z}_\{it}, \hat{c}_\{t}, \\{\hat{z}_\{t-\tau}\\}^L_\{\tau=1})\right)\left|\frac{\partial r_i}{\partial\hat{z}_\{it}}\right|$$
> Thus, the information of $c_t$ is fundamentally present in the noise function $\hat{\epsilon}_\{it}=r_i(\hat{z}_\{it}, \hat{c}_\{t}, \\{\hat{z}_\{t-\tau}\\}^L_\{\tau=1})$.
>
> Third, given the causal structural model in Eq. (4), we build the latent prediction model based on the uncovered latent variables (causal representation) $z_1, \dots, z_T$. This latent prediction model can formulated as $p_\{pred}(\hat{z}_t|\\{\hat{z}_\{t-\tau}\\}^L_\{\tau=1}, \hat{c}_t, \hat{\epsilon}_\{t})$ by using the recovered latent variables $\\{\hat{z}_t\\}^L_\{t=1}$, change factor $\hat{c}_t$, and noise $\hat{\epsilon}_\{t}$. Note that $\hat{c}_t$ is not available at time $t-1$ in prediction mode. One straightforward solution is to build an extra prediction model for change factor $\hat{c}_t$, that is, $p_\{pred}(\hat{c}_t|\hat{c}_\{t-1})$. Interestingly, we can skip this step, since change factor $c_t$ had to be inferred from the latent variables $\\{\hat{z}_\{t-\tau}\\}^L_\{\tau=0}$ as well, like the definition of posterior(encoder) $q_c(\hat{c}_t|\\{\hat{z}_\{t-\tau}\\}^L_\{\tau=0})$. Therefore, we can directly learn the latent predictor via $p_\{pred}(\hat{z}_t|\\{\hat{z}_\{t-\tau}\\}^L_\{\tau=1}, \hat{\epsilon}_\{t})$.
>
> Thanks for pointing this out. We have updated the presentation of the auxiliary latent predictor model and the definition of prior and posterior in VAE structure.

---

> > ### Author Response · Authors · 2022-11-17
> > **response to reviewer AS1z (2)**
> >
> > ***Q3***: The metrics of comparison for the true and estimated latent variables can be much improved. The authors can learn an optimal permutation to compare the latent variables.
> >
> > ***A3***: Thanks for this great point. We would like to note that when computing the MCC between true and estimated latent variables, we have already considered the permutation indeterminacy.
> >
> > As shown in definition 1 and theorem1, we consider the identifiability up to relative minimum indeterminacies. That is, $z_t$  is a component-wise transformation of a permuted version of $\hat{z}_t$. Accordingly, we adopt the following procedure to calculate the MCC. We first apply a nonlinear regression to the recovered factors, aiming to get rid of the component-wise transformation indeterminacy, for each possible pair of the estimated factor and the true one. Then, we calculate all pairs of correlation coefficients (the absolute values of the Spearman’s rank correlation coefficients) between ground-truth latent factors and the estimated latent factors (after the component-wise transformation). We further solve a linear sum assignment problem to assign each latent component to the ground-truth component that best correlates with it, thus finding the correspondence between the estimated factors and the true ones in the latent space. A high MCC means one successfully recovered the true latent factors, up to invertible, component-wise transformation and permutation.
> >
> > We have provided a more detailed description of  MCC calculation in both the main context and appendix.
> >
> > ***Q4***: The invertibility of the transmission plays an important role in the proposed model: the model seems to be under an assumption that the transition is invertible. Please discuss the situation where the transition and change transition functions are not invertible. This would seemingly lead to errors in the inference.
> >
> >
> > ***A4***: Thanks for asking this interesting question. First, let us emphasize that we do not make the assumption for the invertibility of the transition function in our identifiability analysis. (If there is any misunderstanding, please kindly let us know.)
> >
> > Second, we suspect that the reviewer’s concern may be due to our description of the transition prior model in section 3. It is about the recoverability of the noise term given the effect and its parents, instead of the invertibility of the temporal causal process. As we described in Q1/A1, to support more general non-parametric transition process, we formulate the transition prior probability as the product of noise probability and the determinant of the Jacobian matrix by applying the change of variables formula,
> > $$p\left(\hat{z}_\{it}|  \\{\hat{z}_\{t-\tau}\\}^L_\{\tau=1}, \hat{c}_\{t}\right) = p_\{\epsilon_\{it}}\left(r_i(\hat{z}_\{it}, \hat{c}_\{t}, \{\hat{z}_\{t-\tau}\}^L_\{\tau=1})\right)\left|\frac{\partial r_i}{\partial\hat{z}_\{it}}\right|$$
> > Then, we explicitly model the noise function as $\hat{\epsilon}_\{it}=r_i(\hat{z}_\{it}, \hat{c}_\{t}, \\{\hat{z}_\{t-\tau}\\}^L_\{\tau=1})$, which is implemented by the separate MLP network. Then, the noise can be formulated as a function, whose input is target node $\hat{z}_\{it}$, parents node $\\{\hat{z}_\{t-\tau}\\}^L_\{\tau=1}$ and change factor $\hat{c}_\{t}$.
> >
> > Therefore, it means the mapping: $\hat{\epsilon}_\{it} \rightarrow \hat{z}_\{it}$ is **invertible condition on parents nodes $\\{\hat{z}_\{t-\tau}\\}^L_\{\tau=1}$ and change factor $\hat{c}_\{t}$**, where $\hat{z}_\{it}, \hat{\epsilon}_\{it} \in R$ and $\hat{z}_\{t-\tau} \in R^{h_z}$, $\hat{c}_\{t} \in R^\{h_c}$, not **the transition prior function $z_\{it} = f_i(\\{z_\{t-\tau}\\}^L_\{\tau=1}, c_t, \epsilon_\{it})$ is invertible**. Here, we have $\hat{z}_\{it}, \hat{\epsilon}_\{it} \in R$ and $\hat{z}_\{t-\tau} \in R\^{h_z}$, $\hat{c}_\{t} \in R^\{h_c}$, and $h_z$ and $h_c$ denote the dimension of latent variable $\hat{z}_t$ and change factor $\hat{c}_t$.
> >
> > It is worth noting that conditioning set $\\{\hat{z}_\{t-\tau}\\}^L_\{\tau=1}, \hat{c}_\{t}$ has a higher dimension compared to the scalar-to-scalar mapping of $\hat{\epsilon}_\{it} \rightarrow \hat{z}_\{it}$. Therefore, even the noise term is un-invertible, it is a small probability that the noise samples follow the many-to-one mapping and, at the same time, the noise samples condition on the same value of $\\{\hat{z}_\{t-\tau}\\}^L_\{\tau=1}, \hat{c}_\{t}$.
> >
> > Inspired by your question, we conducted an empirical analysis for the settings of non-invertible noise. Particularly, we consider the following latent generation process with squared noise:
> > $z_\{k,t} = q_k(\\{z_\{t-\tau}\\}) + \frac{1}{b_k(\\{z_\{t-\tau}\\})}\epsilon^2_\{k,t}.$
> > The results are shown in Appendix A2.3.3. We can find that the MCC of NPSSM still achieve 0.90.

---

> > > ### Author Response · Authors · 2022-11-17
> > > **response to reviewer AS1z (3)**
> > >
> > > ***Q5***: In Figure 4, the readers are not able to know the order of the latents in scatter plots, in particular, which one is the latent 2 in the MCC matrix
> > >
> > > ***A5***:  We have taken into account the permutation problem when measuring MCC. As mentioned in A2/Q2,  we calculate all pairs of correlation coefficients (the absolute values of the Spearman’s rank correlation coefficients) between ground-truth latent factors and estimated latent factors. We further solve a linear sum assignment problem to assign each latent component to the ground-truth component that best correlates with it, thus reversing any permutations in the latent space.
> > >
> > >
> > > ***Q6***: Some notations in equations need clarifying, e.g., ‘i’ and ‘j’ in Eq 2 need to be clearer, the difference between z_t and \hat z_t in Eq 7 should be clarified, the notations in A.3.2, and so on. Grammar: page2, paragraph 2, line 3, ‘, In particular’.
> > >
> > > ***A6***: Thanks for your comments. We have given the definition of i and j in the 2nd sentence, paragraph 1 of section 2.1, they denote the variable element indices in the initial version. We add the definition of $z_t$ and $\hat{z}_t$ in the 3rd sentence, 1st paragraph of section 3 and fixed the typo. Please see the updated version, your feedback is appreciated.

---

> ### Author Response · Authors · 2022-11-28
> **response to reviewer AS1z**
>
> Dear reviewer AS1z
>
> Thanks for your great efforts in reviewing our paper. We have provided answers to your questions and revised the paper following your suggestions. Please kindly let us know if our response has addressed your concerns.
>
> Thanks again for your valuable review. Looking forward to your reply

---

> ### Author Response · Authors · 2022-12-01
> **response to reviewer AS1z**
>
> Dear reviewer AS1z,
>
> We kindly remind you that it has been 2 weeks since we have submitted the response. So please kindly let us know if our response has addressed your concerns. We will be happy to deal with any additional issues/questions.
>
> Thanks again for your time and effort for providing valuable review. Looking forward to your reply

---

> ### Author Response · Authors · 2022-12-04
> **Discussion period will end soon**
>
> Dear reviewer AS1z,
>
> Hope you are well!  Since the discussion period will end soon, we are very much looking forward to your feedback on our response despite your busy schedule.  It will be highly appreciated if you let us know whether your previous concerns have been properly addressed.  Thank you once again.
>
> We made every effort to address the concerns as you suggested:
> 1. We ***updated the presentation*** of the auxiliary latent predictor model and the definition of prior and posterior in VAE structure to ***clarify the utility of change factors***.
>
> 2. We ***provided a more detailed description of MCC calculation*** in both the main context and appendix.
>
> 3. We ***clarified the possible misunderstanding*** of the assumption for the invertibility of the transition function and discussed the recoverability of the noise term. Inspired by your question, we ***conducted a new empirical analysis*** for the settings of non-invertible noise.
>
> Thanks again for your dedication to reviewing our paper and we are looking forward to your feedback.

---

> ### Comment · Reviewer_AS1z · 2022-12-05
> **Thanks to the authors**
>
> Thanks to the authors for their detailed revisions and explanations. They have addressed all my concerns, and I have raised my score to reflect this.

---

> > ### Author Response · Authors · 2022-12-05
> > **Thanks again for your time and efforts**
> >
> > Dear reviewer AS1z,
> >
> > Thank you again for your feedback. We really appreciate your efforts in reviewing our work and response, and updating your score.

---

### Author Response · Authors · 2022-11-17
**Summary of  Our Revisions**

We sincerely thank all the reviewers for their valuable comments and insightful suggestions.

The reviewers generally have positive opinions of our paper that ***“The work is interesting***” (AS1z), that “The use of functional causal modeling in SSM is ***interesting and promising***”(Uowm), that “The issues of dealing with non-additive noises and non-stationary transition functions are also ***important***,”(Uowm), that “The introduction and the motivation of the work are ***clear***”(AS1z), and that “The paper has ***balanced contributions in theoretical, methodological and empirical aspects***.”(qCMD)


The reviewers also raised constructive and insightful concerns. We made all efforts to address all the concerns by providing additional empirical evidence and improving clarification of our manuscript. We have highlighted the update in red for both main content and appendix, and here is the summary of revisions:

1. **Clarification of technique details and evaluation metric (AS1z, Uowm):** We update our presentation of estimation framework in Section 3, clarify how change variables been used in transition prior model (AS1z),  the relationship between latent causal model and auxiliary latent prediction model (Uowm), and the overall objective function (Uowm). We also clarify how we estimate MCC between the estimated latents and the truth (AS1z, Uowm).

2. **Baselines, new datasets, multi-step forecasting setting and evaluation metric (Uowm, qCMD):** We enlarge our comparison by adding 3 more deep SSM baselines (Uowm, qCMD), 3 more real-world dataset (qCMD) and the new multi-step forecasting settings (qCMD). Besides, we add new evaluation metric CRPS to consider all quantiles. We make every efforts to complete these experiments according to the reviewers’ suggestion. We find that our proposed model still archives the best performance on these extended experiments.

3. **Ablation study and Empirical study of violation assumption settings:** Apart from the ablation of N-NPSSM v.s. NPSSM in the initial version (qCMD), we added the ablation for the transition prior model to verify its effectiveness (Uowm). We also added more empirical studies for violation assumption settings, like the non-invertible noise function (AS1z), instantaneous effect in the latent process (qCMD).

The valuable comments from reviewers are very helpful for us to revise the paper. Your feedback for our revised paper and individual responses are appreciated.

---

### Decision · Program_Chairs · 2023-01-20

**Decision:**

Reject

**Justification For Why Not Higher Score:**

N/A

**Justification For Why Not Lower Score:**

N/A

**Metareview: Summary, Strengths And Weaknesses:**

The paper introduces a state-space model for time series modeling inspired by the general structural causal models.

Strength:
- The method does theoretically make sense and tries to propose a solution to an important problem.
- The theoretical results after the discussion period and clarifications are correct and useful in understanding the proposed method.

Weaknesses:
- Unfortunately, the paper misses connecting their work to the most important developments from the sequence modeling community which is Structural State-Space models which share so much with the work of authors. There is a full body of related works that must be discussed in this paper both theoretically and experimentally. Here is a non-exhaustive list of works on SSMs as a powerful sequence modeling tool that has recently gotten popular:

[1] Gu, A., Goel, K., & Ré, C. (2021). Efficiently modeling long sequences with structured state spaces. arXiv preprint arXiv:2111.00396

[2] Gupta, A. (2022). Diagonal State Spaces are as Effective as Structured State Spaces. arXiv preprint arXiv:2203.14343.

[3] Gu, A., Johnson, I., Goel, K., Saab, K., Dao, T., Rudra, A., & Ré, C. (2021). Combining recurrent, convolutional, and continuous-time models with linear state space layers. Advances in neural information processing systems, 34, 572-585.

[4] Hasani, R., Lechner, M., Amini, A., Liebenwein, L., Tschaikowski, M., Teschl, G., & Rus, D. (2021). Closed-form continuous-depth models. arXiv preprint arXiv:2106.13898.

[5] Gu, A., Dao, T., Ermon, S., Rudra, A., & Ré, C. (2020). Hippo: Recurrent memory with optimal polynomial projections. Advances in Neural Information Processing Systems, 33, 1474-1487.

[6] Smith, J. T., Warrington, A., & Linderman, S. W. (2022). Simplified state space layers for sequence modeling. arXiv preprint arXiv:2208.04933.

[7] Hasani, R., Lechner, M., Wang, T.H., Chahine, M., Amini, A. and Rus, D., 2022. Liquid structural state-space models. arXiv preprint arXiv:2209.12951.

[8] Gu, A., Gupta, A., Goel, K. and Ré, C., 2022. On the parameterization and initialization of diagonal state space models. arXiv preprint arXiv:2206.11893.

[9] Gu, A., Johnson, I., Timalsina, A., Rudra, A. and Ré, C., 2022. How to train your hippo: State space models with generalized orthogonal basis projections. arXiv preprint arXiv:2206.12037

These works not only are the SOTA in sequence and time-series modeling but also directly relate to this paper. Including these frameworks in the paper would require a major revision and is a necessary condition for the acceptance of the paper.


- The authors violated the anonymity condition when providing their code. The identity of the authors is revealed in the git logs, unfortunately.


**Summary Of Ac-Reviewer Meeting:**

We did not have a meeting for this paper because we had a great discussion period and collected enough data to make a decision on this paper.